# 3D multiparametric ultrasound imaging of steatotic liver disease in a study with male rats

Donghyun Lee[1,2,7], Jinseok Heo [2,3,7], Hyeonji Mun [2,4,7], Donghyeon Oh [1,2] ✉, Yongjoo Ahn [1,2,4] ✉ & Chulhong Kim [1,2,3,4,5,6] ✉

Steatotic liver disease (SLD) is the most prevalent chronic liver disease worldwide, requiring precise diagnosis and ongoing monitoring. Ultrasound imaging (USI) is widely used to diagnose steatosis, but it suffers from limited sensitivity/specificity and/or high observer variability. Assessment of hepatic vascular changes during SLD progression has diagnostic relevance because steatosis can induce sinusoidal microvascular dysfunction. Here, we demonstrate 3D multiparametric hepatic USI of SLD in living male rats by coordinating ultrafast Doppler imaging (UFD) and quantitative USI. Vessel volume, perfusion, branching, and tortuosity are quantified from 3D UFD, while tissue attenuation level and heterogeneity are estimated through quantitative USI. Over 8 weeks, hepatic vascular alterations were quantitatively monitored in vivo via 3D UFD during the progression and recovery periods of the SLD. Changes in vascular indices were strongly correlated with hepatic fat accumulation revealed by histopathology, demonstrating great diagnostic potential. Finally, all-around US score comprising five US indices showed a Pearson's coefficient of 0.96 and an average balanced accuracy of 92%, exceeding the performance of each US index alone. Although the translatability to human liver imaging remains limited, these findings suggest that 3D multiparametric USI could serve as a valuable modality for the diagnosis and monitoring of SLD.

Steatotic liver disease (SLD, commonly called fatty liver disease) is a global health concern affecting more than 30% of world's population[1,2]. Due to lifestyle changes and increasing rates of obesity and diabetes, the prevalence of SLD has continued to rise[3], with estimates exceeding 40% in the United States and China[4,5]. SLD is characterized by hepatic steatosis, an excessive fat accumulation in the liver cells[6]. While simple steatosis in early SLD is asymptomatic, it can progress to steatohepatitis, cirrhosis, and even hepatocellular carcinoma[7–9]. This disease's

progression becomes gradually irreversible, with devastating mortality and morbidity[10–12]. In the absence of fibrosis, the degree of steatosis can be a useful marker of SLD progression, so early detection and continuous monitoring of steatosis is vital for appropriate prevention and prompt medical intervention[13].

Typical methods for diagnosing steatosis include liver biopsy, blood tests, magnetic resonance imaging (MRI), and ultrasound imaging (USI). Although a pathological examination is definitive, the

[1]Departments of Convergence IT Engineering, Pohang University of Science and Technology (POSTECH), Pohang, Republic of Korea. [2]Medical Device Innovation Center, Pohang University of Science and Technology (POSTECH), Pohang, Republic of Korea. [3]Departments of Electrical Engineering, Pohang University of Science and Technology (POSTECH), Pohang, Republic of Korea. [4]Departments of Medical Science and Engineering, Pohang University of Science and Technology (POSTECH), Pohang, Republic of Korea. [5]Departments of Mechanical Engineering, Pohang University of Science and Technology (POSTECH), Pohang, Republic of Korea. [6]Opticho Inc, Pohang, Republic of Korea. [7]These authors contributed equally: Donghyun Lee, Jinseok Heo, Hyeonji Mun. ✉e-mail: lumin9219@postech.ac.kr; ahnyj@postech.ac.kr; chulhong@postech.edu

biopsy is highly invasive and suffers from sampling bias[14]. Blood tests can be considered as a minimally invasive alternative, but they are less accurate, not specific to steatosis, and sensitive to external factors[15]. MRI can non-invasively and accurately quantify the fat contents of the entire liver[16,17]. However, its high cost and requisite specialized equipment/facility restrict its accessibility. Due to its availability, non-invasiveness, and cost-effectiveness, USI is the first-line imaging modality for steatosis screening[18], but simple B-mode USI suffers from observer variability and low sensitivity/specificity[19]. Advanced USI techniques, such as attenuation imaging (ATI) and acoustic structure quantification (ASQ), can leverage tissue acoustics to enhance sensitivity/specificity. However, their diagnostic performance and imaging range are inferior to that of MRI[13,20–23]. These drawbacks stem from USI's lack of tissue contrast/resolution and from the indirect measurement of fat accumulation within a limited liver region. Given these considerations, alternatives to quantitative USI for investigating other hepatic alterations beyond tissue characterization would benefit the diagnosis and monitoring of SLD.

Since the liver is a richly vascularized organ, SLD affects the hepatic vasculature in multiple ways. Steatosis can cause endothelial dysfunction, disrupting the balance between vasoconstriction and vasodilation and thereby contributing to elevated intrahepatic vascular resistance[24–29]. Sinusoidal compression can also occur, where hepatocytes enlarged by fatty deposits physically compress the sinusoids[29]. Such sinusoidal changes lead to impaired microcirculation and hepatic blood flow[28]. However, vascular indices remain little considered in diagnosing SLD because conventional Doppler USI is not sensitive enough to detect the microvasculature, imaging only thick vessels such as the hepatic arteries, portal veins, and hepatic veins[30–33]. Ultrafast Doppler imaging (UFD) is an advanced Doppler USI that greatly increases the vascular detection sensitivity by pushing computational speed to the technical limits[34,35]. UFD can visualize and quantify the overall vasculature of organs, from micron- to macro-scales[36–41]. However, recent studies employing hepatic UFD have been constrained to 2D[42,43], which is inadequate for representing the delicate 3D hepatic vasculature. Achieving 3D hepatic UFD would enable imaging of precise vascular features and serve as a complementary modality to reveal hepatic vasculature alterations by SLD, providing further vascular insights and enhanced imaging range.

Here, we demonstrate in vivo 3D multiparametric hepatic USI by integrating UFD and quantitative USI (ATI and ASQ). Steatotic rats were monitored over 8 weeks, validating the SLD progression revealed by histopathology. Vascular rarefaction and distortion during the SLD progression and vascular restoration during the SLD recovery were revealed by four quantitative UFD indices: vessel volume, perfusion, branching, and tortuosity. UFD indices exhibited strong correlations with histopathological hepatic fat content, showing excellent diagnostic performances. Ultimately, integration of the UFD, ATI, and ASQ indices into support vector regression (SVR) models yielded a Pearson's coefficient of 0.96 and a balanced accuracy of 92%, suggesting that our approach can enhance USI's diagnostic accuracy for steatosis. All acronyms used in this study are listed in Supplementary Table 1.

## Results
### 3D UFD and quantitative USI
The configuration of the comprehensive 3D hepatic USI system is illustrated in Fig. 1a. An 18 MHz US probe (L22-14vX, Vermon, France), driven by a data acquisition (DAQ) system (Vantage 256, Verasonics, USA), was utilized to acquire US data transversely in the rat liver. At a single imaging location on the rat's abdomen, respiration was traced in real-time via B-mode interframe errors, automatically triggering an 11-angle planewave US transmission (−10°–10°) to construct an in-phase/quadrature (IQ) frame (Fig. 1b, Supplementary Figs. 1 and 2a, b, and Supplementary Video 1). A total of 600 IQ frames were acquired at a frame rate of 1000 Hz. This imaging procedure was iterated by

traversing the US probe over a scanning range to capture volumetric US data (see "Methods").

The acquired IQ frames were post-processed into power Doppler (PD) images to visualize and quantify hepatic vasculatures (Fig. 1c). Among 600 frames, 150 consecutive stable frames were selected as the ensemble set, and the original PD image was obtained through UFD processing (Supplementary Fig. 2c–h and Supplementary Video 1). Through vascular modeling of successive 2D PD images along the scanning direction, vessel-segmented 3D PD and 3D vessel skeleton images were generated (Supplementary Fig. 3 and Supplementary Video 2). Four quantitative indices were calculated to evaluate the hepatic vasculature. The vessel volume occupancy (VVO) and fractional moving blood volume (FMBV) are hemodynamic indices representing vessel volume and perfusion, respectively, while the vessel bifurcation density (VBD) and sum of angles metric (SOAM) are vascular-morphological indices indicating degrees of vessel branching and tortuosity (See "Methods").

The ATI and ASQ data were also processed from the planewave B-mode US images. ATI estimates the acoustic attenuation associated with hepatic fat accumulation, grounded in the knowledge that steatotic liver tissue loses more acoustic energy than normal tissue (Fig. 1d). Within the region of interest (ROI), the power spectra along the depth windows were extracted, and the estimated attenuation coefficient (EAC) was derived from the linear regression of the natural logarithm of the normalized frequency power ratio. Separately, ASQ was performed to analyze the liver tissue heterogeneity assessed by the US signal's amplitude distribution (Fig. 1e). With increasing hepatic fat contents, the distribution becomes increasingly homogenous, obscuring small liver structures, such as vessels. The deviation from a Rayleigh distribution was quantified using the key distribution parameter $C_m^2$, and the focal disturbance ratio (FDR) was defined based on distribution histograms. Typically, the FDR decreases as hepatic steatosis progresses (See "Methods").

### In vivo 3D hepatic vascular UFD
Figure 2 shows 2D/3D images of the hepatic vasculature of a rat, obtained in vivo via UFD processing. In the B-mode US image, structures other than the abdominal wall are difficult to recognize (Fig. 2a). The major hepatic vessels, including hepatic arteries, portal veins, and hepatic veins, appear hypoechoic; however, they are also challenging to discern. Conversely, the corresponding PD image prominently visualizes the hepatic structures. The vasculatures are clearly defined, enabling precise differentiation of the liver lobe boundaries. It is even possible to distinguish the right median lobe, left median lobe, left lateral lobe, and ventral caudate lobe in a single plane, and to see the vessels connecting the center and periphery of each lobe. In the Gaussian fitting of the line profile (black dotted line), the measured full-width at half-maximum of each peak indicates adjacent vessel diameters of 181 and 112 μm, respectively (Fig. 2b).

The left lateral lobe was mainly targeted for 3D UFD because this region, less influenced by other organs, can be stably imaged in the elevational direction (Supplementary Fig. 4). In the caudal view, a dense vascular network is present with abundant peri-lobular vessels, comprising interlobular arteries, interlobular veins, and sublobular veins (Fig. 2c). Notably, the peri-lobular vessels emanating from the intersecting major hepatic vessel are pronounced, showing the transversal connectivity (magnified view in Fig. 2c). In the cranial view, the detailed ramifications from the central major hepatic vessels to the peripheral vessels are visible stereoscopically (Fig. 2d). Generally, because the portal veins and hepatic arteries are aligned in tandem, the major hepatic vessels can be classified based on whether the major hepatic vessels overlap or not, while the hepatic veins are isolated and larger in diameter (magnified view in Fig. 2d). The interlobular arteries and veins are branched from the hepatic artery and portal vein, while the sublobular veins are branched into the hepatic vein.

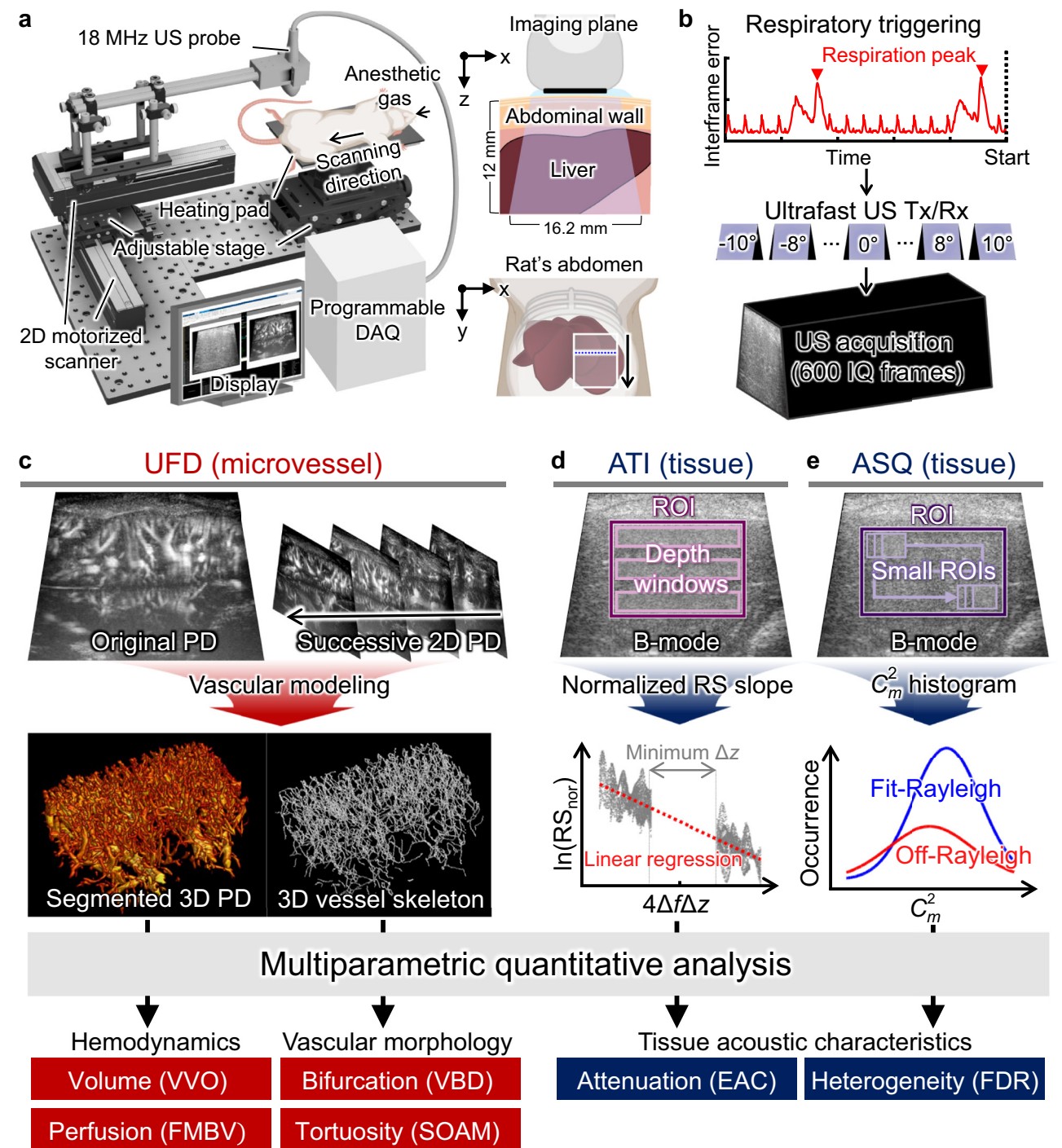

**Fig. 1 | 3D multiparametric hepatic USI: ultrafast Doppler (UFD), attenuation imaging (ATI), and acoustic structure quantification (ASQ). a** Experimental schematic of 3D USI. **b** US data acquisition procedure with automated respiratory triggering, ultrafast ultrasound transmission/reception, and planewave acquisition. **c** 3D UFD processing and quantitative UFD indices: vessel volume (VVO), fractional moving blood volume (FMBV), vessel bifurcation density (VBD), and sum of angles metric (SOAM). **d** ATI processing and estimated attenuation coefficient (EAC) calculation. **e** ASQ processing and focal disturbance ratio (FDR) calculation. DAQ data acquisition, PD power Doppler, ROI region of interest, RS ratio of power spectrum values, $RS_{nor}$ normalized RS, $\Delta f$ frequency step, $\Delta z$ depth gap, and $C_m^2$ key distribution parameter. Rat schematic in (**a**) was created in BioRender. Ahn, M. (2025) https://BioRender.com/yz7h2m4.

## 3D multiparametric USI monitoring of SLDs in live rats

To investigate the hepatic vascular alterations during SLD progression, we ultrasonically monitored normal ($n = 4$) and SLD-conditioned rats ($n = 4$) for 8 weeks (Supplementary Fig. 5a). Separately, we confirmed the temporal changes of intrahepatic fat deposits and microvascular structures in SLD status on a total of 25 rats, divided into five groups ($n = 5$ per group), based on induced-SLD weeks 0, 1, 2, 3, and 4

(Supplementary Fig. 5b). In both experimental schemes, SLD was induced using a methionine-choline deficient (MCD) diet. 3D multiparametric hepatic USI, including the UFD, ATI, and ASQ, was regularly performed, and liver function was tested as necessary via alanine aminotransferase (ALT), aspartate aminotransferase (AST), and total bilirubin (TBIL) measurements. After the final imaging procedure, the rats were sacrificed for histopathology using Oil Red O (ORO), CD31, and

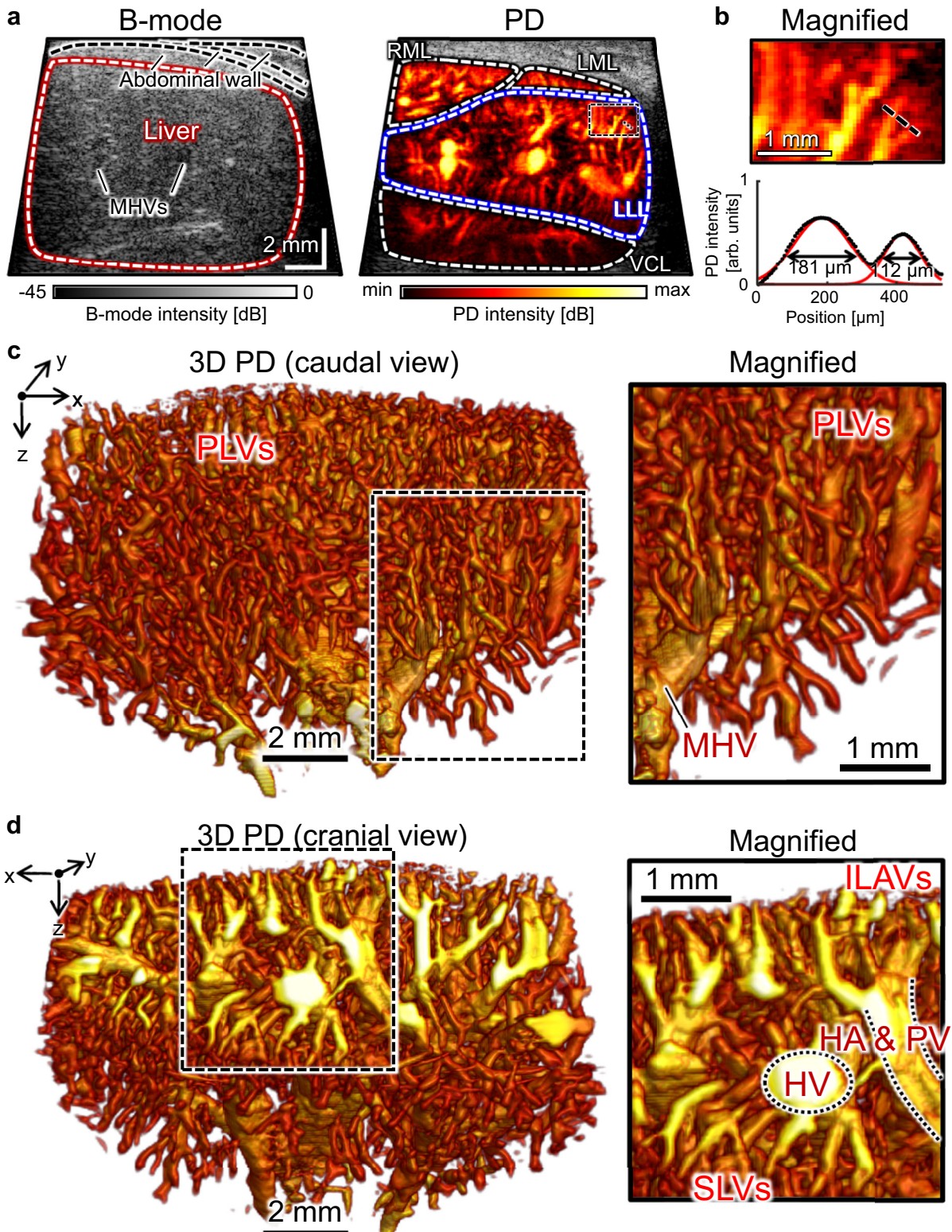

**Fig. 2 | In vivo 3D UFD image of hepatic vasculatures. a** B-mode US image of the rat liver and the corresponding power Doppler (PD) image. RML right median lobe, LML left median lobe, LLL left lateral lobe, and VCL ventral caudate lobe. **b** Finest resolution using Gaussian fitting. **c** Caudal view of 3D PD image of LLL and its magnified view from the black-dotted region. **d** Cranial view and its magnified image. PLVs peri-lobular vessels, MHV major hepatic vessel, ILAVs interlobular arteries and veins, HA hepatic artery, PV portal vein, HV hepatic vein, and SLVs sublobular veins. Source data are provided as a Source Data file.

hematoxylin and eosin (H&E) staining. Hepatic fat accumulation and microvascular damage were quantified by ORO and CD31 staining, respectively, while cellular morphologies were qualitatively assessed using H&E staining. Steatosis grades were classified depending on the

fat percentage: 0–5% (grade 0), 5–33% (grade 1), 33–66% (grade 2), and 66–100% (grade 3) (See "Methods").

Figure 3a–c show representative 3D PD images and cross-sectional overlaid 2D B-mode US and PD images, ordered by steatosis grade and

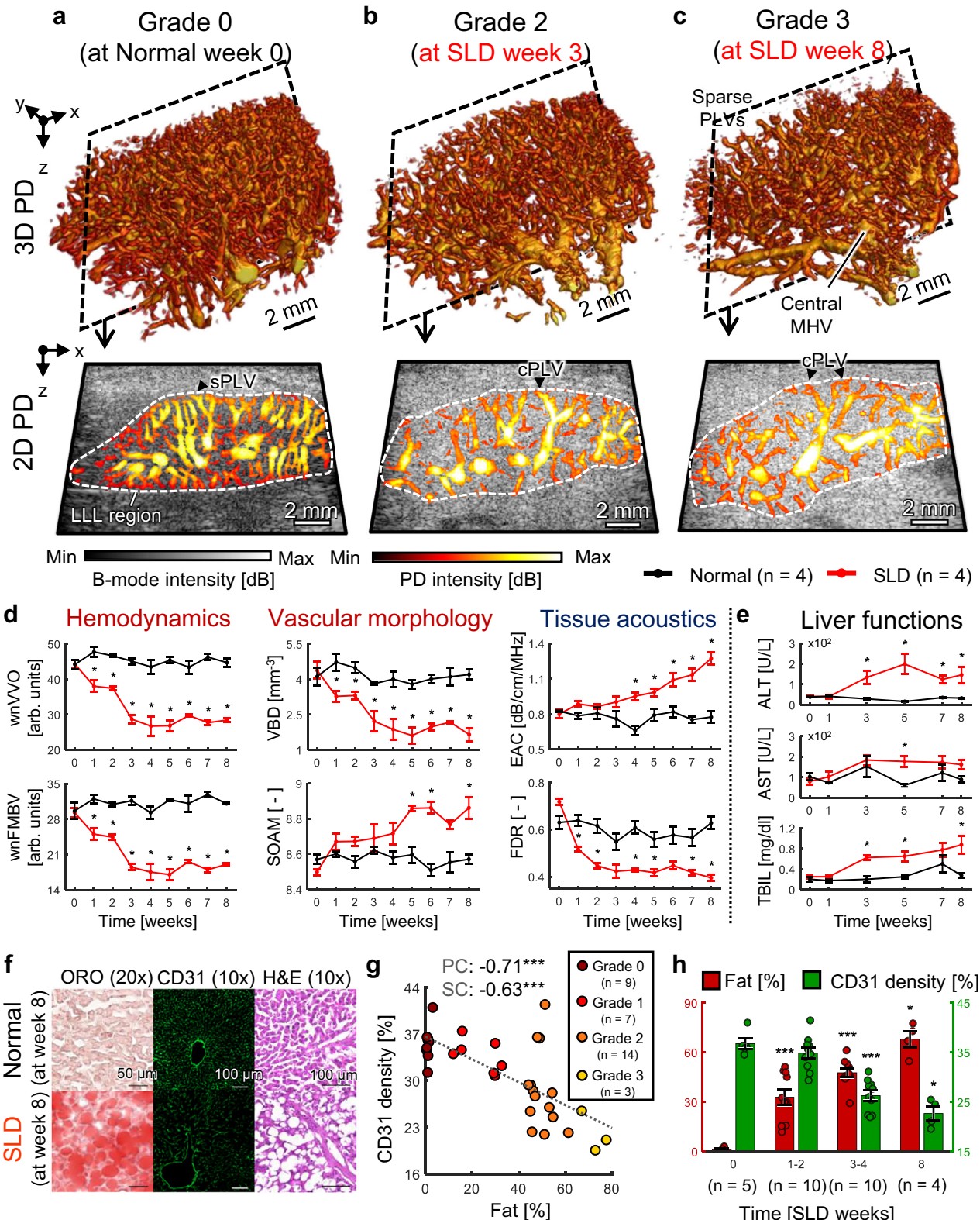

weeks of SLD induction. In the volumetric PD images, compared to grade 0, the vasculatures become increasingly disheveled in grades 2 and 3, with sparser peri-lobular vessels, highlighting the central major hepatic vessels. This vascular trend is clearly observed in the 2D PD images, where predominantly straight peri-lobular vessels are seen in the grade 0 image, but curved peri-lobular vessels appear in grades 2 and 3. Additionally, the B-mode US intensity of the liver region becomes relatively hyperechoic in grade 2 and 3 images.

Differences in hepatic vasculatures are gradually seen over the SLD progression monitoring period. In the normal group, no significant vascular changes are observed over time, whereas the SLD-conditioned rats reveal progressive overall vascular changes, featuring distorted peri-lobular vessels and rarefied vessel distribution (Supplementary Fig. 6a, b).

The normal rats gained weight as they grew naturally, reducing the blood flow detection sensitivity of UFD and subsequently

**Fig. 3 | 3D Multiparametric USI monitoring of SLD in live rats. a–c** 3D PD images and 2D overlaid B-mode US-PD images based on the steatosis grade and SLD-induced weeks. LLL left lateral lobe, PLVs peri-lobular vessels, MHV major hepatic vessel, sPLV straight peri-lobular vessel, and cPLV curved peri-lobular vessel. **d** Monitoring of quantitative US indices over 8 weeks: UFD indices (weight-normalized vessel volume occupancy (wnVVO), weight-normalized fractional moving blood volume (wnFMBV), vessel bifurcation density (VBD), and sum of angles metric (SOAM)), ATI index (estimated attenuation coefficient (EAC)), and ASQ index (focal disturbance ratio (FDR)). Two-sided Mann–Whitney test without adjustment for multiple comparisons at each week (normal vs SLD); *$p < 0.05$. **e** Monitoring of blood biomarkers for liver function tests over 8 weeks. ALT alanine aminotransferase, AST aspartate transaminase, and TBIL total bilirubin. Two-sided Mann–Whitney test without adjustment for multiple comparisons at each week (normal vs SLD); *$p < 0.05$. **f** Oil Red O (ORO), CD31, and hematoxylin and eosin (H&E) stained images for histopathological evaluation in the SLD progression monitoring. CD31 signals are shown in green. **g** Correlation plot between the hepatic fat accumulation and CD31 density. Two-sided Pearson's and Spearman's correlation tests were performed (***$p < 0.001$). Pearson's coefficient (PC) and Spearman's coefficient (SC) represents linear and non-linear monotonic relationships. **h** Quantitative histopathological results from the SLD progression monitoring and validation schemes at SLD-induced weeks 0, 1–2, 3–4, and 8. Two-sided Mann–Whitney test between time points (comparisons shown only vs. week 0); *$p < 0.05$, ***$p < 0.001$. All statistical results, including other comparisons and exact $p$-values, are provided in Supplementary Tables 2 and 3. All data are presented as mean ± SEM. Source data are provided as a Source Data file.

decreasing the hemodynamic indices (Supplementary Fig. 7). To account for this, VVO and FMBV were adjusted for weight, resulting in weight-normalized VVO and FMBV (wnVVO and wnFMBV).

Over the 8-week monitoring, all US indices relatively remain constant in the normal rats, but the SLD-conditioned rats showed changes from the baseline values (Fig. 3d). The wnVVO, wnFMBV, and VBD values steadily decrease until week 5 and stabilized thereafter, whereas the SOAM and EAC consistently increases over the 8 weeks. The FDR declines rapidly by week 3, and then slightly decreases by week 8. These variations suggest that hepatic fat accumulation may affect not only microscopic vessels at the sinusoidal level, but also affect more macroscopic vessels, supporting the hepatic vasculature changes observed in UFD imaging.

Overall, blood biomarker levels from in vitro blood serum tests were higher in the SLD group than the normal group (Fig. 3e). The ALT and AST levels, indicators of hepatocellular damage, and the TBIL level, associated with liver metabolism and excretory function, remain consistently elevated in the SLD group, following the measurements at week 1. These biomarkers indicate impaired liver function in the SLD group, confirming progression to a pathologic state.

Histopathological changes became progressively evident (Fig. 3f and Supplementary Fig. 8). In the SLD at week 8, significantly larger red droplet areas, reduced green structures, and expanded hepatic cells with inflammatory infiltrations appear respectively in the ORO, CD31, and H&E images. The correlation between the fat percentage and CD31 density has a strong Pearson's coefficient (PC) and a significant Spearman's coefficient (SC) (Fig. 3g). This inverse relationship validates that hepatic fat accumulation is associated with impairment of the sinusoidal structures.

The differences in the fat percentage and CD31 density across induced-SLD weeks 0, 1–2, 3–4, and 8 reveal a definitive longitudinal trend (Fig. 3h). The hepatic fat accumulation incrementally increases with induced-SLD weeks, while the CD31 density shows no significant changes between weeks 0 and 1–2, but gradually decreases at weeks 3–4 and 8. These changes in CD31 density may validate the changes in UFD indices caused by the SLD, considering the liver's homeostasis. The CD31 density may specifically correlate with hemodynamic indices, related to the blood flow into and out of the microvascular structures. The decrease in hemodynamic indices with a corresponding maintenance in CD31 density at weeks 1 and 2, as well as the decline in CD31 density without a decrease in hemodynamic indices at week 8 compared to week 4, is speculated to result from compensatory homeostatic mechanisms of the micro- and macro-vasculatures. These temporal tendencies suggest that while the SLD progression may be gradual, changes could vary across different indices.

Additionally, to validate the changes observed in our USI findings, we performed LYVE1 staining on a subset of normal and SLD rats. In contrast to CD31, LYVE1 density shows a marked decline at week 2, followed by a more gradual decrease at week 4 (Supplementary Fig. 9). Given that a decrease in LYVE1 may indicate endothelial dysfunction[44–46], the temporal patterns of LYVE1 and CD31

density at weeks 0, 2, and 4 suggest that structural alterations in the sinusoids occur subsequent to initial functional impairment. At week 8, LYVE1 levels remain comparable to those at week 4, still indicating impaired sinusoidal function, which may have contributed to the reduction in CD31 observed at week 8. Changes in UFD indices may reflect both functional and structural alterations in the microvascular architecture. All statistical analyses are summarized in Supplementary Tables 2 and 3.

### 3D multiparametric USI monitoring of SLD recovery in live rats

As shown in Supplementary Fig. 10, the progress of the SLD was monitored to assess observable recovery and potential repair of the impaired hepatic vasculature. Four rats were monitored for 8 weeks: the MCD diet was switched to normal chow after week 4 of the imaging and vitamin E was prescribed to improve histopathologic outcomes[47–49]. Blood collection and rat sacrifice were also conducted (See "Methods").

In the 3D/2D PD images, from week 0 to week 4, similar trends occur in the SLD progression period (Fig. 4a, b vs Fig. 3a, b). Then the cluttered and tortuous vascular distribution seen at week 4 is restored by week 8, becoming comparable to the original condition (Fig. 4c). This restoration is well demonstrated in the 2D PD images, where the curved peri-lobular vessels straighten out and become compact and straight (Supplementary Fig. 11). In the B-mode US, the liver region intensity reverts to its original echogenic level by week 8.

Six US indices and blood biomarkers show patterns consistent with the results of SLD progression monitoring through week 4 (Fig. 4d, e vs Fig. 3d, e). However, during the subsequent recovery period, weeks 5 through 8, these measures reverse their trends, returning close to each baseline value from week 0 by week 8.

In microscopic staining images, compared to the SLD-conditioned rats at week 4, the recovery-conditioned rats at week 8 exhibit decreased red fat content in ORO, restored green microvasculature in CD31, and cellular expansion in H&E, resembling the normal rats at week 0 (Fig. 4f). The quantified histological results show clear recovery trends (Fig. 4g). At week 8, the recovery group demonstrates reduced hepatic fat accumulation and increased CD31 density compared at induced-SLD weeks 3–4, confirming the recovery of hepatic alterations.

However, the influence of SLD remains in the recovery group (Fig. 4h). Compared to the normal group at week 8, the fat percentage in the recovery group at week 8 is 4.8% higher and the CD31 density is 3.8% lower. The reduced FDR at week 8 may be attributed to residual hepatic fat, while the incomplete restoration of wnVVO, wnFMBV, and VBD at week 8 may be linked with the lower CD31 density. These findings suggest that the effects of SLD could be long-lasting, and they demonstrate that recovery from SLD process can be observed using 3D multiparametric USI.

As with SLD progression monitoring, LYVE1 staining was also performed in the recovery group for additional validation. In contrast to CD31, LYVE1 density at week 8 returned to levels comparable to those of the normal group at weeks 0 and 8 (Supplementary Fig. 12). This suggests that although functional recovery of the microvasculature has

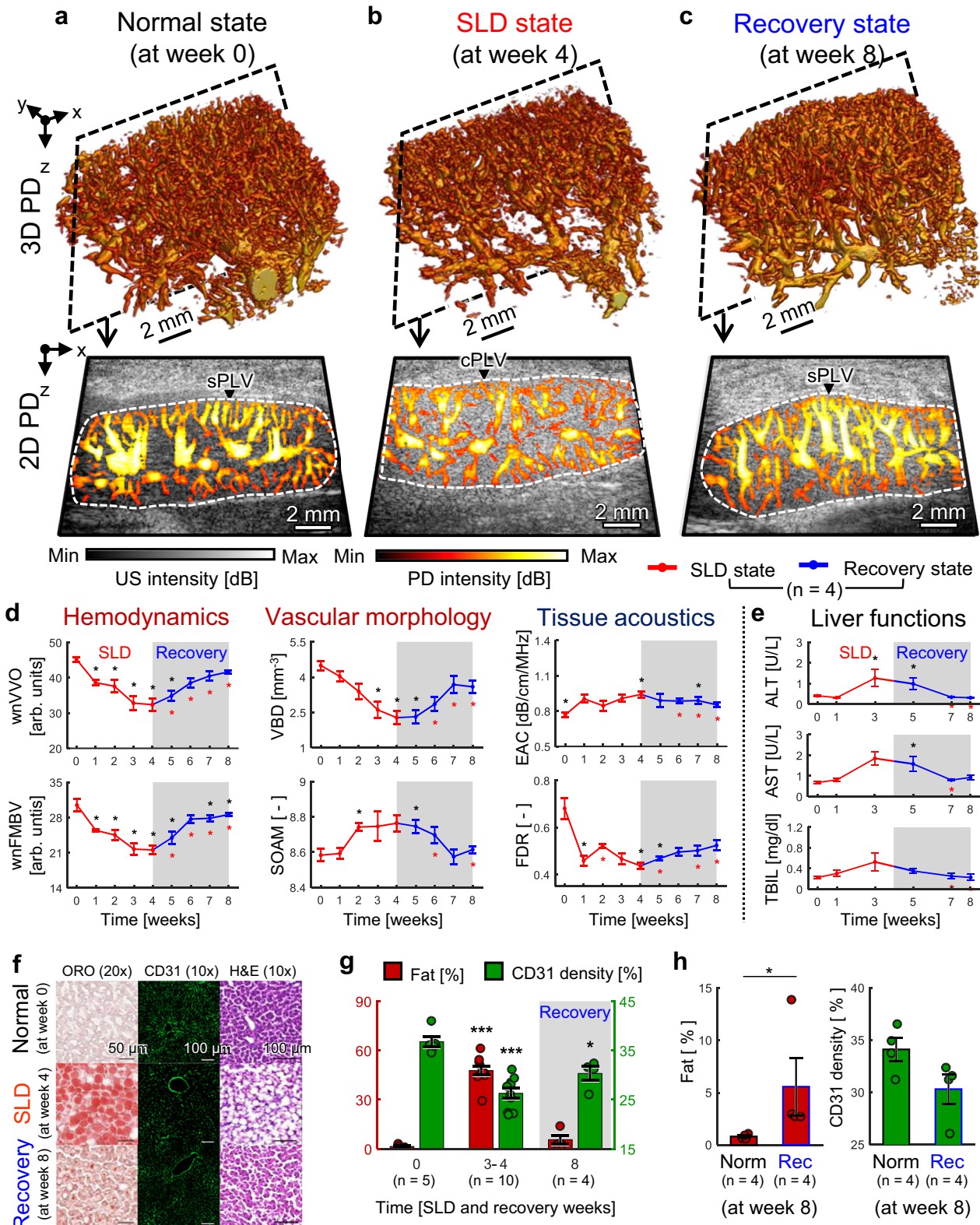

occurred, structural restoration remains incomplete, further supporting the possibility that these structural changes may be irreversible. All statistical analyses are summarized in Supplementary Tables 4 and 5.

**Multiparametric analyses for evaluating the diagnostic performance of 3D USI**

To explore the potential of 3D UFD indices as practical diagnostic markers, we analyzed the correlation between the histopathological and 3D USI data, calculating PC and SC for a total of 37 datasets. The diagnostic performance was further evaluated by the receiver operating characteristic (ROC) curve. Classifications were conducted for steatosis grades 0 vs. 1–3, 0–1 vs. 1–2, 0 vs. 1, and 1 vs. 2, while grades 0–2 vs. 3 and 2 vs. 3 were excluded due to insufficient data for grade 3 ($n = 3$). Along with the empirical ROC, the analytical ROC and 95% confidence interval were estimated, with the sensitivity, specificity, and balanced accuracy of classification

**Fig. 4 | 3D Multiparametric USI monitoring of SLD recovery in rat livers.** 3D PD and 2D overlaid B-mode US-PD images acquired at **a** week 0, **b** week 4 (SLD progression), and **c** week 8 (recovery). **d** Monitoring of quantitative US indices over 8 weeks: UFD indices (weight-normalized vessel volume occupancy (wnVVO), weight-normalized fractional moving blood volume (wnFMBV), vessel bifurcation density (VBD), and sum of angles metric (SOAM)), ATI index (estimated attenuation coefficient (EAC)), and ASQ index (focal disturbance ratio (FDR)). **e** Monitoring of blood biomarkers for liver function tests over 8 weeks. ALT alanine aminotransferase, AST aspartate transaminase, and TBIL total bilirubin. Two-sided Mann–Whitney test without adjustment for multiple comparisons at each week in (**d**) and (**e**); recovery vs normal (Fig. 4d vs. Fig. 3d); recovery vs SLD (Fig. 4e vs. Fig. 3e); *$p$ < 0.05. Black and red asterisks denote comparisons with the normal and red

SLD groups, respectively. **f** Oil Red O (ORO), CD31, and hematoxylin and eosin (H&E) stained images for histopathological evaluation in the SLD progression and recovery monitoring. CD31 signals are shown in green. **g** Quantitative histopathological results from the SLD progression and recovery monitoring and validation schemes at SLD-induced weeks 0, 3–4, and 8. Two-sided Mann–Whitney test without adjustment for multiple comparisons between time points (comparisons shown only vs. week 0); *$p$ < 0.05, ***$p$ < 0.001. **h** Comparison of the fat percentage and CD 31 density values between the recovery (Rec) and normal (Norm) groups at week 8. Two-sided Mann–Whitney test without adjustment for multiple comparisons; *$p$ < 0.05. All statistical results, including other comparisons and exact $p$-values, are provided in Supplementary Tables 4 and 5. All data are presented as mean ± SEM. Source data are provided as a Source Data file.

determined at the optimal point on the analytical ROC (see "Methods").

In the correlation analysis, the hemodynamic indices demonstrate the highest correlation, followed by the vessel morphological indices, while the tissue acoustic indices exhibit the lowest correlation, with PC and SC magnitudes above 0.9, 0.8, and 0.7, respectively (Fig. 5a). These findings support the hypothesis that changes in UFD indices are substantially related to hepatic fat accumulation, and they suggest that UFD indices could be utilized as diagnostic biomarkers.

The same support is also evident in the ROC analyses (Supplementary Fig. 13). In classifying grade 0 vs. 1–3, wnVVO an wnFMBV exhibit excellent balanced accuracy (above 90%). VBD and SOAM show comparable balanced accuracies to those of EAC and FDR, respectively. This pattern is amplified in the grade 0–1 vs. 2–3 classification. Notably, wnVVO and wnFMBV show remarkable balanced accuracy (above 95%), with SOAM showing higher balanced accuracy than FDR. For grade 0 vs.1, except for wnFMBV, the UFD indices fall short of FDR yet exceed EAC, while in the grade 1 vs. 2 classification, all UFD indices demonstrate better performance than FDR, and only VBD perform below the levels of EAC.

Compared to the 3D UFD indices, the 2D UFD indices, taken as the median of five large-parenchymal frames, show similar performances in the wnVVO and wnFMBV measures, but their performance as indices of VBD and SOAM is inferior (Supplementary Fig. 14). One explanation could be that 3D UFD provides the vascular connectivity across multiple 2D cross-sections, offering a comprehensive representation of vessel morphology. Furthermore, 2D UFD indices are prone to location-dependent variability, which further supports the superiority of the 3D UFD indices (Supplementary Fig. 15).

Finally, we integrated the UFD, ATI, and ASQ indices to produce an all-around US score to estimate the hepatic fat accumulation. Utilizing an SVR algorithm, we tested all possible combinations of the six US indices, a total of 63 cases (Supplementary Table 3). To mitigate overfitting, we employed a nested Monte Carlo cross-validation approach to select the optimal combination using a separate dataset. The performance of SVR models is evaluated by PC, and this process identified a combination of five indices (wnVVO, wnFMBV, VBD, SOAM, and EAC) as the optimal subset (Supplementary Fig. 16a).

The combination of five indices (wnVVO, wnFMBV, VBD, SOAM, and EAC) exhibits a PC of 0.96 and a SC of 0.95 and demonstrates an average balanced accuracy of 92% in all classifications, which exceeds that of all US indices (Fig. 5b). Although the performance improvements within the SVR models are modest due to the excellent individual capabilities of wnVVO and wnFMBV, considering the variability of the indices, this multi-index interactive all-around US score would provide more consistent performance across various measurement conditions.

Additionally, to evaluate the model's generalization performance, we calculated the average test set performance across all possible combinations in each iteration of the nested Monte Carlo cross-validation (Supplementary Fig. 16b). Numerous combinations yielded PC values exceeding 0.9. Moreover, as the number of combined

indices increased, performance became more stable, indicating robust generalization. These findings confirm the robustness of our approach and demonstrate that UFD indices can be harmonized with conventional US indices (See "Methods").

## Discussion

In this paper, we establish a 3D multiparametric hepatic USI system by combining UFD with ASQ and ATI. Utilizing this system, we elucidate the rat's in vivo 3D hepatic vasculatures down to a diameter of 113 μm, and we confirm the presence of vascular alterations during SLD progression. Furthermore, by conducting correlation and ROC curve analyses, we find that changes in UFD indices (wnVVO, wnFMBV, VND, and SOAM) are strongly associated with intrahepatic fat accumulation, showing excellent diagnostic performances. Finally, we incorporate the five US indices via SVR to construct robust all-around US score. The resulting 3D multiparametric hepatic USI can enable accurate diagnosis and continuous monitoring of SLD, promoting timely medical intervention and medication response measurement.

This study is distinctive in providing the observation and quantification of the overall vascular alterations associated with SLD in such a detailed and longitudinal fashion. Several technical approaches introduced to enhance imaging robustness enabled successful long-term volumetric scan observations. First, volume scanning was optimized using real-time US-based respiratory gating, which enabled US acquisition exclusively during cardiac cycle intervals, facilitating efficient 3D Ultrafast US acquisition. Second, by strategically applying various techniques (debiasing, intensity normalization, non-local means filtering, histogram equalization, and Jerman filtering) at each relevant stage, a stable vascular modeling framework was established, enabling consistent and accurate volumetric image processing and quantification. Indeed, our UFD indices demonstrated excellent inter-individual reproducibility and inter-system agreement, reflecting overall high reproducibility (Supplementary Figs. 17 and 18). This robustness enabled the stable execution of the approximately 133 imaging procedures required for the study, including SLD progression monitoring, SLD validation, and SLD recovery monitoring, with a minimum of 72, 25, and 36 procedures, respectively (Supplementary Figs. 5 and 10).

Unlike conventional Doppler US, which is limited to observing phenomena such as portal hypertension and decreased hepatic artery resistivity in large hepatic vessels, UFD can visualize and quantify the comprehensive hepatic vascular system in 3D, encompassing both microvascular and macrovascular structures. This ability enables the detection of vascular alterations that emerge in the early stages of SLD, which may enhance the diagnostic utility. Notably, our 3D UFD approach, with enhanced morphological assessment, demonstrated that SLD progression is associated with integrated vascular alterations.

Over an 8-week SLD progression monitoring, gradual vascular rarefaction and torsion are revealed by increasing wnVVO, wnFMBV, and VBD, along with decreasing SOAM. As the induced-SLD weeks progress, the fat percentage increases, while the CD31 and LYVE1 densities decrease, suggesting that the hepatic fat accumulation leads to microvasculature disruptions. We speculate that, as demonstrated

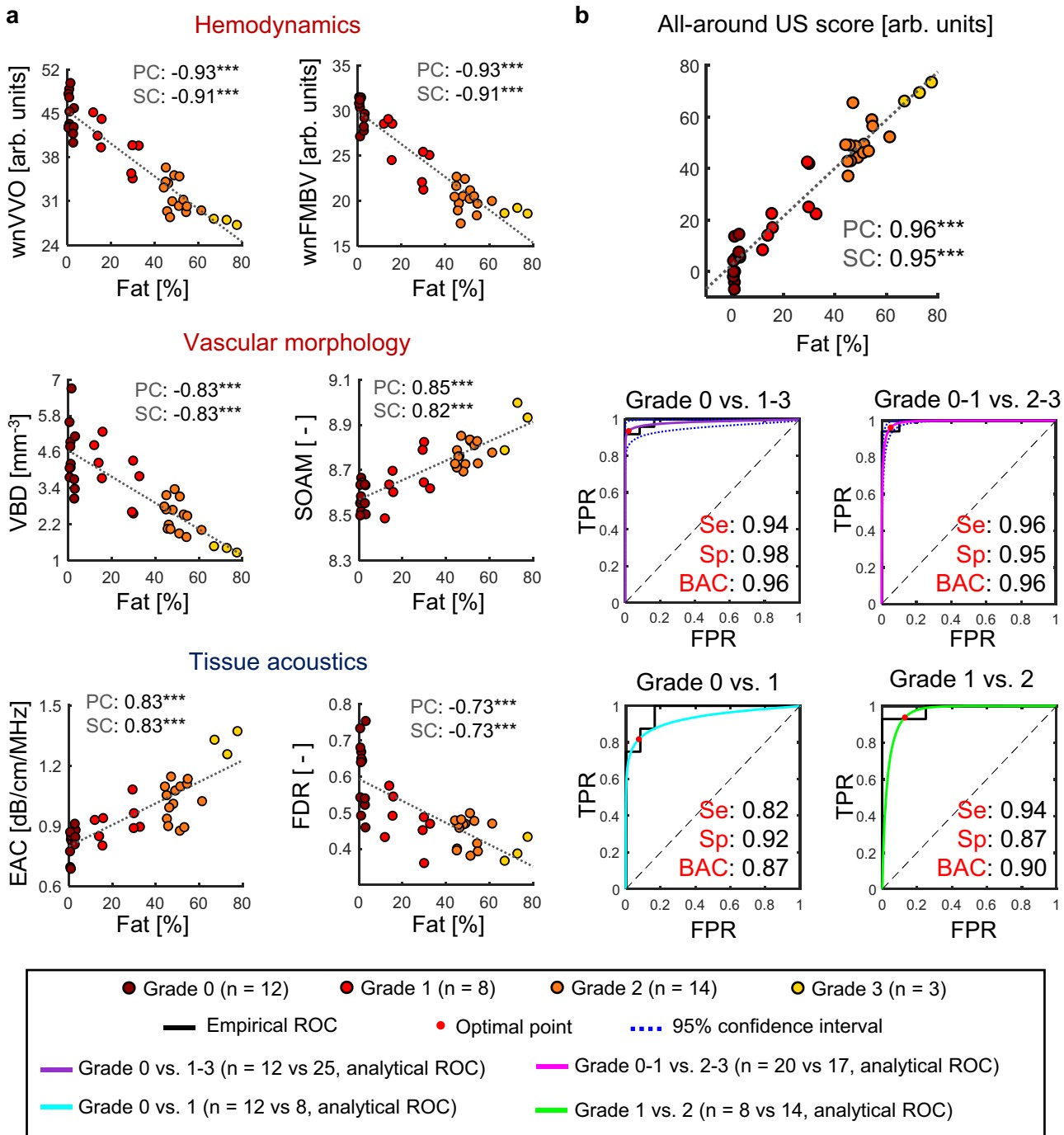

**Fig. 5 | Multiparmetric analyses and all-around US score using a support vector regression (SVR) approach. a** Scatter plots of showing correlations between the fat percentage and 3D USI indices (*n* = 37 for each index): UFD indices (weight-normalized vessel volume occupancy (wnVVO), weight-normalized fractional moving blood volume (wnFMBV), vessel bifurcation density (VBD), and sum of angles metric (SOAM)), ATI index (estimated attenuation coefficient (EAC)), and ASQ index (focal disturbance ratio (FDR)). Two-sided Pearson's and Spearman's correlation tests were performed (****p* < 0.001). **b** All-around US score intergrating five 3D USI indices (wnVVO, wnFMBV, VBD, SOAM, and EAC) using SVR (*n* = 37).

Two-sided Pearson's and Spearman's correlation tests were performed (****p* < 0.001). In the classifications of grade 0 vs. 1 and grade 1 vs. 2, estimating 95% confidence intervals is excluded due to insufficient data. PC Pearson's coefficient, SC Spearman's coefficient, ROC receiver operating characteristic, TPR true positive rate, FPR false positive rate, Se sensitivity, Sp specificity, and BAC balanced accuracy. The results for each index were derived only once from each subject (i.e., biological replicate). Exact *p*-values are provided in source data. Source data are provided as a Source Data file.

in previous studies[24–29], this may result from endothelial dysfunction, a functional alteration occurring in the early stages of SLD that leads to structural changes in the microvasculature. Lipid droplets from hepatocytes hypertrophied by the fat accumulation physically compressing the sinusoidal structure, namely sinusoidal compression, may also have contributed to this[28,29]. Meanwhile, the observation of

vascular rarefaction in the hemodynamic indices (wnVVO and wnFMBV) is contrary to findings from previous micro-CT studies[50]. This contradiction probably arises because the hemodynamic indices assess actual perfusion rather than the vasculature itself, with similar findings reported in previous cirrhosis studies[42]. The same level of CD31 density between weeks 0 and 1–2 of our SLD monitoring

experiments may explain this opposing relationship. Liver homeostasis might maintain the CD31 level to compensate for microvascular damage caused by hepatic fat accumulation. However, the actual perfusion may have been reduced due to microvascular impairment.

We also conducted SLD recovery monitoring, demonstrating that the SLD recovery process is observable in our 3D USI. This capability would be highly valuable for testing SLD treatments, particularly given the currently limited number of FDA-approved treatments. The UFD indices showed incomplete recovery, suggesting that overall in vivo blood flow within the hepatic vascular system may not have been fully restored. In the temporal trend of LYVE1 and CD31 density, LYVE1 returned to near-normal levels at recovery week 8, whereas CD31 levels remained below normal. This indicates that, despite functional recovery of the sinusoids, the structural alterations in the microvasculature induced during SLD progression may be irreversible.

The UFD indices demonstrate better performance than ATI and ASQ indices in correlation and ROC curve analyses, highlighting the potential of UFD indices to predict the SLD status. We would like to emphasize that UFD indices reflect not merely the presence of blood vessels, but the in-vivo blood flow within the integrated vascular system, thereby representing an effective vascularity index from the perspective of vascular dynamics. UFD indices may provide a more comprehensive reflection of SLD progression than CD31 density, which could account for their higher correlation coefficients with fat percentage. While the UFD indices perform well independently, the 3D USI conditions were optimized for the UFD. Compared to the 10 cm thickness of the human liver, the rat liver is diminutive, approximately 1 cm, and high-frequency US signals were utilized for acquisition to achieve fine resolution. Such experimental conditions might make the analysis of tissue acoustic characteristics challenging.

The all-around US score, derived using multivariate SVR, demonstrate that the UFD indices are complementary to conventional US indices used in clinical practice, supporting the hypothesis that UFD indices can provide an alternative perspective compared to conventional ones. Since USI is often utilized as a dual-modality technique complementary to MRI, photoacoustic imaging, computed tomography, and positron emission tomography, integrating UFD with these modalities can be expected to further enhance functionality[51–53].

The primary objective of this study was to explore the association between SLD progression, reflected by hepatic fat accumulation, and vascular alterations across multiple scales, encompassing both microvascular and macrovascular domains. Accordingly, the study focused on investigating the complementary roles of UFD and quantitative USI, rather than conducting a direct comparison between them or aiming for precise hepatic fat quantification. In this context, quantitative USI was intended to serve as a complementary modality to UFD, enabling non-invasive confirmation of hepatic fat accumulation during SLD monitoring without reliance on histological analysis.

Among various quantitative USI techniques, two widely utilized tools for SLD diagnosis, ATI and ASQ, were employed in this study. Both techniques are well-established, supported by several clinical research, and implemented in commercially available US systems[21,22,54,55]. While other quantitative indices such as the backscatter coefficient, speed of sound, and envelope statistics are indeed valuable[56–58], they were excluded due to methodological and practical considerations. Specifically, backscatter coefficient estimation requires precise calibration with reference phantoms, and speed of sound estimation requires specialized acquisition sequences for spatially uncompounded data at multiple steering angles, both of which were challenging under our experimental condition. Furthermore, improving the accuracy of the speed of sound estimation technique requires access to raw channel data, which is generally unavailable in clinical US systems and, when combined with the aforementioned requirements, substantially increases computational complexity. Other envelope statistics, such as Nakagami and normalized local

variance imaging, differ from ASQ in their statistical models and mathematical formulations, yet share the fundamental principle of characterizing the statistical distribution of echo amplitude patterns. From the perspective of research efficiency, including all techniques would have imposed substantial additional burden with limited incremental benefit.

Our findings suggest that integrating UFD with quantitative USI parameters enhances the accuracy of predicting SLD development. Unlike conventional quantitative USI techniques that reflect tissue acoustic properties, UFD provides complementary information on hepatic vascularity. Recent studies have explored the use of deep learning models to accurately quantify hepatic fat by integrating multiple quantitative USI indices[59,60]. Vascular indices derived from UFD may offer valuable complementary features for AI-based fat quantification, providing information distinct from that captured by conventional quantitative USI indices.

Our study has several limitations. First, MCD diet-induced SLD only partially recapitulates human SLD pathology. Unlike human SLD, which is characterized by systemic metabolic dysfunction (obesity, insulin resistance, hyperlipidemia, etc.), MCD diet–induced SLD is confined to hepatic metabolic impairment[61–65]. This distinction introduces an inherent degree of uncertainty regarding the translational relevance of its pathological findings with regard to the inflammation and fibrosis observed in humans. Second, the number of data samples is limited. Although nested Monte Carlo cross-validation was performed to verify the generalization ability of the SVR models, the risk of data overfitting remains. Securing a larger dataset is essential for future utilization. Third, due to the resolution limitations (~100 μm) of our current imaging system, detecting sinusoidal capillarization, which requires visualization of fenestrations (~0.1 μm)[66], is challenging. The onset of sinusoidal capillarization in rats with SLD-induced by the MCD diet varies across studies[25,27,29,67]. Sinusoidal capillarization is associated with the progression of SLD, as well as with inflammation and fibrosis, making it a key component of human SLD pathophysiology that warrants consideration in future studies.

Extending our research to extensive clinical studies could address these limitations. In practice, we are currently preparing an extended study on human liver imaging, utilizing the Vantage 256 system along with a convex-array (GE C1-6-D) and linear-array (GE 9L-D) probes to acquire high-quality human liver images (Supplementary Fig. 19a, b). A total of 400 frames (500 Hz frame rate) and 240 frames (300 Hz frame rate) were acquired via the intercostal window using the convex-array and linear-array probes, respectively, from a single healthy volunteer (male, 20–30 years old). Ensemble frames for UFD were selected at 0.8-s intervals, considering the human cardiac cycle and resting heart rate (60–100 bpm). As shown in the figures, the human hepatic vasculature is visualized in sufficient detail across an extensive region with the convex-array probe, and in finer detail within a narrower region with the linear-array probe.

However, technical challenges must still be carefully considered and overcome. For effective monitoring and medical intervention in near real-time, the UFD procedure requires accelerated data processing through GPU-based computation. Further, although our rat-model study was conducted in 3D, obtaining 3D images from the human abdomen is challenging. Using matrix probes or the fan-shaped tilt scanning (e.g., the Wobbler mechanism) may enable the acquisition of reliable volumetric images even within a limited acoustic window[40,68–70]. Although 2D image acquisition may be considered, both 2D and 3D imaging would require motion compensation to account for respiration[71,72]. In human abdominal imaging, low-frequency US probes with a central frequency of approximately 3 MHz are typically used, and the imaging depth will be substantially greater than in rat imaging. This difference can reduce the spatial resolution and pulse repetition frequency, potentially compromising UFD image quality. However, given the differences in vascular scale

and heart rate between humans and rats[73,74], a reduced pulse repetition frequency may still provide a sufficient ensemble length to compensate for potential degradation in image quality. The human liver, located deep within the abdomen, can be thick enough to impede US beam transmission, necessitating techniques such as pulse compression in UFD[75,76]. High time-bandwidth product pulses, such as coded excitation, which has already been implemented in various US applications, can enhance the signal-to-noise ratio at greater imaging depths by mitigating the trade-off between frequency and penetration, thereby enabling the use of linear transducers with higher center frequencies. Additionally, given the complexity of the liver vasculature, color Doppler techniques, which can specify the blood flow direction and velocity, would be highly functional. Ultimately, by building on technical advances, integrating UFD into clinical practice could expand its applications to surgical procedures such as liver transplantation, in addition to diagnosis and monitoring of various diseases, such as hepatitis B, cirrhosis, and hepatocellular carcinoma.

## Methods

### 3D USI
Utilizing a DAQ-driven USI system (Vantage 256, Verasonics, WA, USA), in vivo liver imaging was non-invasively performed on an anesthetized rat. A high-frequency linear-array transducer (L22-14vX, Vermon, France; center frequency, 18.0 MHz; bandwidth, 44%) was coupled with this system to acquire US data. Prior to US acquisition, the rat's respiration was monitored using the real-time B-mode interframe difference to exclude respiratory intervals from the imaging procedure (Supplementary Video 1). The acquisition was commenced following the second respiratory peak, determined by the threshold. At a pulse repetition frequency of 20 kHz, 11-angle planewave US pulses (0°, ±2°, ±4°, ±6°, ±8°, ±10°) were transmitted to reconstruct a single US frame (Supplementary Fig. 1). The acquired raw radiofrequency (RF) data were immediately processed into IQ data, resulting in a B-mode frame rate of 1000 Hz. After acquiring a total of 600 US frames at a single imaging location, the transducer was sequentially moved 0.2 mm to the next location by a 2D motorized stage controlled via DAQ triggering. By reiterating this procedure along the scanning range (8–12 mm), 3D volumetric data were obtained. Based on the transducer specifications, a dense scanning step size of 0.2 mm was set to better visualize vascular network connectivity and enable more precise computation of vascular morphological indices (Supplementary Note 1). The imaging procedure, including US acquisition and image reconstruction, required 10 s per single location, resulting in a total imaging time of 10–12 min. The imaging field-of-view (FOV) was specified as a trapezoidal shape (top width, 12.8 mm; bottom width, 16.2–16.8 mm; and depth, 12–14 mm), providing a lateral view that is wider than the probe foot print. These FOVs were adjusted in depth to maximize the liver inclusion, and the corresponding bottom widths were automatically adjusted.

### Respiration triggering, ensemble selection, and Doppler processing
Prior to DAQ, the rat's respiratory motion was monitored in real-time via the B-mode interframe error at a framerate of 50 Hz (Supplementary Video 1). The interframe error was computed using the root-mean-square of pixel-by-pixel differences in successive B-mode intensities (Supplementary Fig. 2a). This error was then smoothed using a moving average, and respiratory peaks were detected (MATLAB R2022a, Mathworks, MA, USA; function *movmean* and *findpeaks*) (Supplementary Fig. 2b). The system was automatically switched to US acquisition mode 0.2 s after detecting the second peak. The time delay for initiating acquisition and the threshold for peak detection were adjusted based on each experimental animal's respiration rate and motion intensity.

After the planewave US acquisition, 150 frames corresponding to a relatively consistent cardiac phase were selected as an ensemble from the 600 US IQ frames obtained at each location (Supplementary Fig. 2c–e and Supplementary Video 2). The interframe error was calculated in the same manner as for respiration triggering, but at a higher frame rate of 1000 Hz, without smoothing. To capture diastole flow after ventricular systole, 150 sequential frames avoiding peaks caused by heartbeats, were selected. From the selected IQ ensemble frames, the blood signal was extracted using clutter filtering. Singular value decomposition based clutter filtering was employed to retain the vascular signal from the US signal, which contained tissue, blood, and noise components (Supplementary Fig. 2f). We applied a double threshold method based on the spatial cross-correlation matrix to eliminate the noise signals while maximizing the preservation of blood signals. Out of a total 150 spatiotemporal ranks, the upper threshold was set at 15 and the lower threshold was set at 75; the range between these thresholds was adopted to represent the blood signals. Finally, power averaging and logarithmic compression were performed on these blood signals to generate a PD image in a dB scale (Supplementary Fig. 2g, h).

### Vascular modeling and quantitative UFD indices
In the all UFD images, the imaging target, the left lateral lobe, was first manually segmented using the MATLAB Volume Segmenter with reference to the B-mode and PD images. For equivalent criteria, each segmented imaging set was rescaled to a range of 30 dB from its maximum 1% average intensity within the left lateral lobe region label. For quantitative analysis, a vessel segmentation process was then performed to distinguish the vessel structures from other regions inside the left lateral lobe. The vessel segmentation process involved the following steps (Supplementary Fig. 3a–d): debiasing to remove the inherent system noise, non-local mean filtering to reduce the speckle effect (available at https://www.bic.mni.mcgill.ca/PersonalCoupepierrick/OBNLMFilter)[77], histogram equalization to enhance vessel contrast, and 2D/3D Jerman filtering to strengthen vessel connectivity (available at https://github.com/timjerman/JermanEnhancementFilter)[78]. Finally, this processed image was combined with the debiased PD image in a 1:1 ratio, and a vessel mask was generated from the combined image through thresholding. Based on this vessel mask, a 2D/3D UFD and a 2D/3D vessel skeleton were constructed. VVO and FMBV were calculated based on the vessel mask and debiased PD image.

$$\text{VVO} = \frac{V_{vessel}}{V_{liver}} \qquad (1)$$

$$\text{FMBV} = \frac{\sum \text{sPD}}{V_{liver}} \qquad (2)$$

The subsequent skeletonization process was also performed based on this mask. In this process, vessel branches with shorter than 10 pixels in length were removed. The branch points for the VBD were computed by the characteristics of the vessel skeleton, and the SOAM calculation was included only if the vessel path length was longer than 5 pixels.

$$\text{VBD} = \frac{N_{bp}}{V_{liver}} \qquad (3)$$

$$\text{SOAM} = \frac{1}{N_b} \sum \frac{\sum \text{angle}}{L_b} \qquad (4)$$

### ATI and ASQ implementation
For ATI and ASQ, a B-mode guided ROI was selected, and the corresponding raw US data was analyzed. We implemented the ATI function based on spectral normalization at different frequencies to analyze the attenuation from the selected ROI[79,80]. An area of the ROI measuring

approximately 8 × 4 mm (width × height) was selected. The window size along the depth direction for the short-time Fourier transform (STFT) was set to ten wavelengths, with an overlap of 97.5%. The STFT was applied along the depth axis for each column. The power spectra from multiple windows at the same depth was then averaged. Then, for each depth $z_k$, the ratio between the power spectrum values at two adjacent frequency bins ($f_{i-1}, f_i$) was calculated:

$$\mathrm{RS}(f_i, z_k) = \frac{S(f_i, z_k)}{S(f_{i-1}, z_k)} \tag{5}$$

where RS is the ratio of the power spectrum values, $f_i$ is a frequency, $z_k$ is a given depth, and $S$ is the power spectrum. The normalized ratio of the power spectrum values at reference depth $z_r$ was calculated as

$$\mathrm{RS}_{\mathrm{nor}}(f_i, z_k, z_r) = \frac{\mathrm{RS}(f_i, z_k)}{\mathrm{RS}(f_i, z_r)} \tag{6}$$

where $\mathrm{RS}_{\mathrm{nor}}$ is the normalized ratio of the power spectrum values and $z_r$ is a reference depth. Because the attenuation is known to follow the equation:

$$\ln\left[\mathrm{RS}_{\mathrm{nor}}(f_i, z_k, z_r)\right] = -4\alpha(f_i - f_{i-1})(z_k - z_r) \tag{7}$$

where $\alpha$ is the attenuation coefficient, $\alpha$ was estimated using the least squares method model implemented in the MATLAB function *fitlm* with the *RobustOpts* option. A consecutive frequency range was selected in which the EAC remained within ±25% of the value estimated at the frequency exhibiting the overall maximal power spectrum. A rectangular ROI was carefully selected to include only the area inside the left lateral lobe, and a total of five reliable US frames, covering large areas of the hepatic parenchyma, were selected to adequately represent tissue attenuation. The median value from the five calculated EACs was used as the representative EAC.

In the ASQ analysis, we selected a rectangular primary ROI from the liver parenchymal area, ensuring it was large enough to represent the echo amplitude distribution while avoiding large vessel structures. Although US signal distribution is theoretically expected to follow the Rayleigh distribution, the actual distribution observed in a healthy liver deviates from this ideal due to the presence of small liver structures, such as vessels. With increasing hepatic fat accumulation, these structures are obscured by the fat contents, leading the distribution to approximate the Rayleigh distribution. The raw US data from the selected ROI was used for further analysis. Multiple secondary ROIs (sROIs), each measuring 0.5 × 0.5 mm (width × height), were sampled from the primary ROI to calculate the local degree of deviation from the Rayleigh distribution across the different regions of the liver parenchyma. The deviation degree was evaluated by key distribution parameters, $C^2$ and $C_m^2$, calculated for each sROI using following equations:

$$C^2 = \frac{\pi}{4 - \pi} \frac{\sigma^2}{\mu^2} \tag{8}$$

$$C_m^2 = \frac{\pi}{4 - \pi} \frac{\sigma_m^2}{\mu_m^2} \tag{9}$$

Here, $\sigma^2$ is a variance, $\mu$ is an average, $\sigma_m^2$ is a modified variance, and $\mu_m$ is a modified average. Modified values were derived by using partial samples of sROI, excluding excessively high values. A sROI where $C^2/C_m^2$ exceeds the threshold is considered to deviate from the theoretical distribution. The ratio of deviated sROIs to those following the distribution is defined as FDR. Five different frames were selected as for the ATI. The mean value from the five calculated FDRs was used as the representative FDR.

## Animal model and experimental schemes

A total of 37 male Wistar rats (Rattus norvegicus, outbred, CrljOr-i:Wistar, 7 weeks old, approximately 200 × g at week 0), obtained from Orient Bio (Seongnam, Korea), were involved in our study. Rats were acclimatized for 1 week before experimentation. Normal rats were provided standard rodent chow (5L79, Charles River Laboratories, Wilmington, MA, USA; 18.4% protein, 5.7% fat, 55.9% carbohydrate by difference), and the SLD state was induced by feeding a methionine–choline-deficient (MCD) diet (A02082002BR, Research Diets Inc., New Brunswick, NJ, USA; 16% protein, 21% fat, 63% carbohydrate). MCD diet depletes phosphatidylcholine precursors, impairing the secretion of very low-density lipoproteins in the liver. In turn, it results in increased fat deposition in the liver.

Male Wistar rats were used in this study, as previous research has demonstrated that they develop the most pronounced steatosis in response to the MCD diet[81]. Sex-based analyses were not performed, as the study included only male animals. All rats were housed under a 12 h light/12 h dark cycle in a temperature- and humidity-controlled facility (22–25 °C, 40–50% humidity), with ad libitum access to the food and water.

Three experimental schemes were designed and conducted: (1) SLD progression monitoring involving normal rats ($n = 4$, normal group) and SLD-conditioned rats ($n = 4$, SLD group), (2) SLD validation using 25 rats divided into five groups ($n = 5$ per group) based on induced-SLD durations of 0 (normal group), 1, 2, 3, and 4 weeks (SLD group), and (3) SLD recovery monitoring involving recovery-conditioned rats ($n = 4$, recovery group) (Supplementary Figs. 5 and 10). The sample size for each experimental scheme was determined using G*Power[82,83], based on findings from previous study[84] (see Supplementary Note 2).

Identical SLD induction periods were assumed to result in the same pathological status (e.g., SLD-conditioned rats at week 3 in the SLD progression monitoring scheme was considered equivalent to SLD group of induced-SLD weeks 3 in the SLD validation scheme). However, the monitoring results were analyzed exclusively within the monitored group (Figs. 3d, e and 4d, e).

In the SLD progression and recovery monitoring schemes, 3D hepatic USI was performed weekly to thoroughly inspect hepatic alterations, and blood samples were collected bi-weekly to alleviate the burden of blood withdrawal. In contrast, in the SLD validation scheme, 3D hepatic USI was carried out only once for each rat to correlate with histological confirmation, with no blood collection. All rats ($n = 37$) were euthanized following the final imaging procedure.

In the SLD recovery monitoring, a gavage of vitamin E (258024-5G, Sigma-Aldrich) was initiated for the recovery-conditioned rats after the imaging procedure of experimental week 4 along with a transition from the MCD diet to standard rodent chow. A solution of 10 mg/ml vitamin E dissolved in water was administered daily by mouth via an oral syringe, at a volume of 10 ml/kg of the body weight. Vitamin E may control the oxidative stress and inflammation induced by an MCD diet. It suppresses lipid accumulation and peroxidation to reduce the liver damage. The combined intervention of discontinuing the MCD diet and administering vitamin E was intended to significantly alleviate hepatic steatosis, facilitating clearer observation of recovery in the SLD recovery group (i.e., recovery-conditioned rats).

## Pre-imaging protocol

The experimental animals were fasted for approximately 12 h prior to the imaging experiment, with water available ad libitum. Anesthesia was initially induced using a preclinical anesthetic gas system (VIP 3000 Veterinary Vaporizer, Midmark, OH, USA) with a flow rate of 1.5 L min⁻¹ and an isoflurane concentration of 3–5% in oxygen. After the rats lost consciousness, anesthesia was maintained at the same flow rate with a 1–2% isoflurane concentration throughout the experiment, using a 3D-printed nose cone. The breathing rate was closely

monitored via real-time B-mode interframe errors, and the isoflurane concentration was adjusted as needed to maintain a respiratory rate of approximately 1 Hz. Abdominal hair was removed using animal clippers and depilatory cream, and the body weight was measured. The animals were secured using medical bandages (Micropore™, 3 M, MN, USA) in a supine position on a thermal heating pad. To minimize shading and artifacts caused by the rib cage and surrounding organs, the probe position and scanning range were carefully adjusted to capture the full cross-section of the left lateral lobe, from the superior part, where the rib cage minimally obscures the view, to the inferior region just before the small intestines appear.

### Blood tests

After imaging, 0.5–1 mL blood samples were collected from the retro-orbital sinus, using heparinized capillary tube (Marienfeld HSU-2900000, Germany). Blood sampling was conducted at weeks 0, 1, 3, 5, 7, and 8, with the left and right sinuses alternated to minimize harm to the animals. To separate the serum, the samples were centrifuged at $3000 \times g$ and 4 °C for 10 min. Serum samples were analyzed for ALT, AST, and TBIL using the DRI-CHEM NX600V automatic analyzer (Fujifilm, Japan) and corresponding test slides (DRI-CHEM SLIDE, Fujifilm, Japan).

### Histopathology

Following the final imaging procedure, the rats were euthanized, perfused with saline, and their livers were collected. The left lateral lobe of each liver was fixed with 10% neutral buffered formalin and incubated in 30% sucrose at 4 °C for 2–3 days. Fixed liver samples were embedded in tissue OCT compound (Tissue-Tek® O.C.T. Compound, SAKURA), and frozen at −80 °C. Specimen blocks were sectioned at a thickness of 10 μm using a cryostat (CM1860, Leica).

Lipid droplets in liver sections were stained with ORO solution (O1391, Sigma-Aldrich) and counterstained using hematoxylin. Stained sections were mounted with VectaMount® AQ Aqueous Mounting Medium (H-5501-60, Vector Laboratories), and images were acquired using a ZEISS Axioscan 7 at 20× magnification. Ten images were randomly selected from a single stained slide, avoiding large vessel holes, and the fat percentage was quantified by thresholding on a 2108 × 1218 pixel ROI of each image (ImageJ, National Institutes of Health, Bethesda, MD, USA). The average of these ten values was used as the representative value. Steatosis grades were categorized based on the fat percentage: 0–5% as no steatosis (grade 0), 5–33% as mild steatosis (grade 1), 33–66% as moderate steatosis (grade 2), and 66–100% as severe steatosis (grade 3)[84,85].

Immunofluorescence staining for CD31 and LYVE1 was performed to label vascular endothelial cells and liver sinusoidal endothelial cells, respectively. After washing with 1× phosphate-buffered saline, blocking was conducted with 10% fetal bovine serum, 1% bovine serum albumin and 0.3% Triton ×-100 in 1× phosphate-buffered saline. Next, the sections were incubated overnight at 4 °C with primary antibodies, anti-CD31 (1:200, AF3628, R&D) and anti-LYVE-1 (1:200, AF7939, R&D), and then with the secondary antibodies, rabbit anti-goat Alexa Fluor 488 (1:2000, Invitrogen) and donkey anti-sheep Alexa Fluor 488 (1:2000, Jackson ImmunoResearch), for 1 h at room temperature. After each antibody incubation, the slides were washed 3 times for 5 min each time. Stained samples were mounted with VectaMount® AQ Aqueous Mounting Medium (H-5501-60, Vector Laboratories). Z-stacked images were obtained using a confocal laser scanning microscope (ZEISS LSM 900 with Airyscan 2) at 10× magnification and processed with ZEISS ZEN 3.8 software to perform maximum intensity projection. Three randomly chosen images with 256 × 256 pixel ROI were quantified for CD31 and LYVE1 densities with ImageJ plugin "Vessel Analysis" and then averaged.

Cryosections were stained with H&E, and images were acquired using the ZEISS Axioscan 7 at 20× magnification.

### ROC curve and diagnostic performance analyses

The analytical ROC and 95% confidence interval were estimated under the assumption that each index follows a normal distribution. The sensitivity and specificity of each index were determined as the optimal point on the analytical ROC curve where Youden's J statistic is maximized. The balanced accuracy, defined as the average of sensitivity and specificity, was used as a representative measure of the diagnostic performance for each index. This ensures a fair assessment of each index by addressing the imbalance of classification classes (steatosis grades 0 vs. 1–3, 0–1 vs. 1–2, 0 vs. 1, and 1 vs. 2) and incorporating the significance of each class.

### SVR and nested Monte Carlo cross-validation

An SVR technique was employed to integrate US indices for approximating the histopathological hepatic fat percentage. SVR was adopted due to its robustness to overfitting and noise, even with limited data, and its strong generalization performance[86]. The SVR was implemented using the MATLAB function *fitrsvm*, with a predefined regularization parameter (set to one) and a linear kernel for simplicity. Since increasing the number of variables in SVR does not guarantee improved performance, all possible combinations of the six US indices, a total of 63 cases, were tested.

Nested Monte Carlo cross-validation was employed to identify the optimal feature subset and assess the model's generalization performance while minimizing the risk of overfitting[87,88]. In the outer loop, all 37 data were divided into a training set (80%) and a test set (20%). Subsequently, fivefold cross-validation was conducted within the inner loop, using only the training set. The performance results obtained using fourfolds for training and onefold for validation were used exclusively to select the optimal combination. Thereafter, the model, trained using the averaged weights and biases from the fivefold cross-validation, was evaluated on the test set in the outer loop, and this performance was used to assess generalization. This process was repeated 100 times; namely, the whole loop was executed 100 times. Finally, a single SVR model was trained using the selected optimal feature combination to enable comparison with the original six US indices.

### Statistical analysis

Group data are presented as mean ± standard error of the mean (SEM). The significance of differences between groups was assessed using the two-sided Mann–Whitney test (also known as Wilcoxon rank-sum test), implemented through the built-in MATLAB function, *ranksum*. This test was chosen due to the small sample size per comparison group ($n = 4$–10), which limits the reliability of normality assumptions. A $p$-value of less than 0.05 was considered statistically significant.

### Human liver imaging

Human liver imaging was performed on a single healthy volunteer (male, 20–30 years old), who was recruited without regard to sex, as this was a preliminary imaging. The volunteer was fully informed about the imaging procedures and provided written informed consent. No financial or other compensation was provided to the volunteer. UFD imaging was conducted using a DAQ-controlled 256-channel research USI platform (Vantage 256, Verasonics, WA, USA) equipped with both convex-array (GE C1-6-D, GE Healthcare, USA; 192 elements; center frequency: 3.75 MHz) and linear-array (GE 9L-D, GE Healthcare, USA; 192 elements; center frequency: 6.2 MHz) transducers. The right intercostal window was used to image the right hepatic lobe.

### Ethical statement

All procedures for experimental animal were reviewed by the Institutional Animal Care and Use Committee of Pohang University of Science and Technology (POSTECH-2023-0099, approved on 21 Aug. 2023) and were strictly followed. Human volunteer experiments were

performed in accordance with a protocol approved by the Institutional Review Board of Pohang University of Science and Technology (POSTECH-PIRB-2023-A001-C2, approved on 17 Jan. 2025).

## Reporting summary

Further information on research design is available in the Nature Portfolio Reporting Summary linked to this article.

## Data availability

The main data supporting the findings of this study are included in the main text, figures, and supplementary information. The representative subset of the US raw data generated in this study has been deposited in the Zenodo database [https://doi.org/10.5281/zenodo.16918233][89]. Due to the large size of the entire raw dataset (~50 TB), only a subset is hosted on Zenodo; however, the complete dataset can be provided upon request to the corresponding author (contact: chulhong@postech.ac.kr). Request will be answered within a month. The data will be shared under a data use agreement permitting use for non-commercial research purpose only and requiring compliance with relevant ethical and privacy regulations. Source data are provided in this paper. Source data are provided with this paper.

## Code availability

All post-processing and quantifications were performed using custom MATLAB-based code. Example MATLAB scripts for the essential analysis steps have been deposited in the Zenodo database [https://doi.org/10.5281/zenodo.16918233][89].

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

## Acknowledgements

This work was supported by the Basic Science Research Program through the National Research Foundation of Korea (NRF), funded by the Ministry of Education (RS-2020-NR049599 received by C.K.); a National Research Foundation of Korea (NRF) grant funded by the Korea government (MSIT) (RS-2023-NR077260 received by C.K.; RS-2024-00411069 received by Y.A.; RS-2024-00335346 received by D.O.);  the Commercialization Promotion Agency for R&D Outcomes (COMPA) funded by the Ministry of Science and ICT (MSIT) (RS-2025-02304660 received by C.K.); a grant of the Korea Health Technology R&D Project through the Korea Health Industry Development Institute (KHIDI), funded by the Ministry of Health & Welfare (RS-2024-00512879 received by C.K.; RS-2024-00438673 received by Y.A.); and by the BK21 FOUR (Fostering Outstanding Universities for Research) project.

## Author contributions

All authors contributed to the conceptualization of the study. D.O., D.L., and J.H. developed the 3D multiparametric USI system, and H.M. conducted the histopathological analysis. D.L., J.H., and H.M. jointly performed the overall experiments. The quantification analysis was led by D.L., with assistance from J.H. and H.M. The project was supervised by C.K., Y.A., and D.O. All authors were involved in discussing the results and writing the manuscript.

## Competing interests

Chulhong Kim has financial interests in OPTICHO, which, however, did not support this work. All the other authors declare no other competing interests.
