## [Transparent Peer review file · Nature Communications]

3D multiparametric ultrasound imaging of steatotic liver disease in a study with male rats

Corresponding Author: Professor Chulhong Kim

Version 0:

Reviewer comments:

Reviewer #1

(Remarks to the Author)
Please see the attached pdf.

(Remarks on code availability)
I highly encourage the authors to consider sharing the code.

Reviewer #2

(Remarks to the Author)
The study showcases 3D multiparametric ultrasound imaging as a tool for diagnosing and monitoring steatotic liver disease in rats. By integrating ultrafast Doppler imaging and quantitative USI, it quantified vascular and tissue changes over disease progression and recovery, showing strong correlation with hepatic fat accumulation. A combined US score achieved superior diagnostic accuracy, highlighting the potential of 3D USI for SLD management. However, limitations in imaging interpretation, and the diet-induced model's relevance to human disease warrant attention in interpreting and generalizing the findings. Addressing these limitations will help refine the utility of 3D UFD imaging and its application to broader experimental or clinical settings. Overall, this is an important study.

Specific comments:

1. The methionine-choline-deficient diet induces SLD but may not fully recapitulate human SLD pathology, particularly in terms of inflammation and fibrosis. Please discuss the relevance and limitation of disease model used.

2. Figure 2: UFD imaging provides a much clearer delineation of liver structures and vascular networks, overcoming the limitations of conventional B-mode ultrasound, which struggles to distinguish hepatic structures or major vessels. The relatively accurate measurement of vessel diameters (e.g., 175 μm and 113 μm) and the stereoscopic visualization of vascular branching, particularly the transversal connectivity of peri-lobular vessels and the ramifications from major hepatic vessels, set an advancement in hepatic vascular imaging. However, there are potential limitations to consider. The left lateral lobe was primarily targeted for 3D UFD imaging due to its stable positioning and minimal interference from other organs. This focus might limit generalizability across other liver lobes, especially regions with greater motion artifacts or proximity to other organs. How can this be resolved?

3. Are there any measurements fluctuations affected by respiratory or cardiac motion in live imaging of rats?

4. Figure 3: Temporal trends in hepatic fat accumulation and microvascular alterations are well-captured. Fat accumulation can alter microvascular architecture and impede blood flow within the liver. However, the precise mechanisms linking the two are not fully elucidated. Comparing Fig. 3d and 3e, vascular changes occur early by 1 week of diet, whereas changes to liver functional markers were more apparent by 3 weeks of diet. How do your findings stand in view of current literature/evidence?

5. Figure 4: Vascular traits appeared responsive to recovery intervention. Do vascular changes reflect genuine structural

restoration or merely less compression from reduced liver fats. For example, vascular branching by VBD, does it represent angiogenic/ active remodeling response? While the study links imaging indices to vascular and hepatic changes, additional molecular analyses in histology (e.g., angiogenic or inflammatory markers) could provide deeper physiological/ biological insights.

6. Lyve1 coupled with CD31 is a better marker for validating liver microvasculatures against your US findings. To resolve sinusoidal capillarization versus compression, can your image reconstruction discern healthy levels of fenestration versus structural alteration happening in capillarization? This would have more direct attributes in relation to what is known in human liver pathology.

7. Figure 5: Diagnostic performance of 3D USI indices was benchmarked against classification grades. In the ROC curves, how do UFD indices perform compared to conventional US indices in sensitivity and specificity?

(Remarks on code availability)

Version 1:

Reviewer comments:

Reviewer #1

(Remarks to the Author)

The authors have done an excellent job addressing the reviewers' comments, and the quality of the work has been significantly improved after the revision. A few additional issues were noted:

1. The issue of excessive use of acronyms has not been resolved. I would highly recommend considering reducing the use of acronyms in this manuscript to improve readability.
2. Line 421: "speed of sound estimation requires specialized acquisition sequences": The same statement is true for ultrafast Doppler.
3. Line 422: "obtaining accurate estimates using these two techniques necessitates access to raw channel data": This statement is incorrect. Backscatter coefficient measurement does not require access to channel data. Some of the speed of sound methods do not require access to channel data either.
4. Line 470: "thereby enabling the use of linear transducers with higher center frequencies": I am unsure if the use of linear transducers would be appropriate for liver imaging, given the limited field of view of linear transducers.
5. In response to the reviewer's question about the sample size, the authors added a post hoc power analysis to the revised manuscript. The value of this post hoc power analysis is unclear. As a general guideline, post hoc power analysis should not be used, as explained in the following publication: Heinsberg LW, Weeks DE. Post hoc power is not informative. *Genet Epidemiol.* 2022 Oct;46(7):390-394.

(Remarks on code availability)

Reviewer #2

(Remarks to the Author)

My comments have been adequately addressed.

(Remarks on code availability)

Table R3. List of Acronyms.

Acronyms	Paraphrase	Acronyms	Paraphrase
SLD	Steatotic liver disease	PLVs	Peri-lobular vessels
USI	Ultrasound imaging	ILAVs	Interlobular arteries and veins
UFD	Ultrafast Doppler imaging	SLVs	Sublobular veins
MRI	Magnetic resonance imaging	MCD	Methionine-choline deficient
ATI	Attenuation imaging	ALT	Alanine aminotransferase
ASQ	Acoustic structure quantification	AST	Aspartate transaminase
HAs	Hepatic arteries	TBIL	Total bilirubin
PVs	Portal veins	ORO	Oil Red O
HVs	Hepatic veins	H&E	Hematoxylin and eosin
SVR	Support vector regression	wnVVO	Weight-normalized VVO
DAQ	Data acquisition	wnFMBV	Weight-normalized FMBV
IQ	In-phase/quadrature	PC	Pearson's coefficient
PD	Power Doppler	SC	Spearman's coefficient
VVO	Vessel volume occupancy	ROC	Receiver operating characteristic
FMBV	Fractional moving blood volume	RF	Radiofrequency
VBD	Vessel bifurcation density	FOV	Field-of-view
SOAM	Sum of angles metric	RMS	Root-mean-square
ROI	Region of interest	SVD	Singular value decomposition
EAC	Estimated attenuation coefficient	sROIs	Secondary ROIs
FDR	Focal disturbance ratio	VLDL	Very low-density lipoproteins
MHV	Major hepatic vessels	NBF	Neutral buffered formalin
RML	Right median lobe	PBS	Phosphate-buffered saline
LML	Left median lobe	FBS	Fetal bovine serum
LLL	Left lateral lobe	BSA	Bovine serum albumin
VCL	Ventral caudate lobe		

Reviewer #1

[Key results]

In this manuscript, the authors demonstrated that 3D multiparametric hepatic ultrasound imaging (USI) could be a useful modality for diagnosis and monitoring of steatotic liver disease (SLD). They performed an animal study on 37 rats that underwent 3D ultrasound liver scans in vivo. The 3D multiparametric USI method used by the authors involves the combination of two well-established USI techniques, ultrafast Doppler imaging (UFD) and quantitative USI. UFD provides visualization of the liver microvasculature, -and quantitative USI provides information about acoustic properties of the liver. It is well known in the literature that quantitative USI parameters such as the attenuation coefficient and acoustic structure quantification are correlated with liver fat content, and this study added knowledge to the literature by showing that UFD-derived parameters were also correlated with liver fat content and might have superior diagnostic performance compared with quantitative USI parameters (i.e., tissue-derived US indices).

Reply:

Let us begin our reply by saying that we sincerely appreciate your review of our manuscript and your encouraging comments. Please find our detailed responses to your specific comments below.

[Validity of conclusions]

The conclusion that “3D multiparametric USI can be an invaluable modality for diagnosis/monitoring of SLD” is well supported by the data presented in the manuscript. However, the claimed superiority of UFD relative to quantitative USI is not adequately supported, for several reasons:

Reply:

Before addressing your subsequent questions, we think that we must clarify the key points of our research to avoid potential misunderstandings.

The core purpose of this research was to explore the correlation between SLD progression and multi-scale (including micro- and macro-scale) vascular alterations. It was not to directly compare UFD and quantitative USI or to precisely quantify hepatic fat percentage. Our findings suggest that the combined use of both UFD and quantitative USI indices enhances the accuracy of predicting SLD development. In the sections below, we provide more detailed responses to your questions.

We also have revised portions of the manuscript to clarify the study’s objective and to prevent potential misinterpretations by readers.

1) The authors only examined two quantitative USI parameters, the attenuation coefficient and the acoustic structure quantification. Other quantitative USI parameters that are known to strongly correlate with liver steatosis, such as the backscatter coefficient (<https://doi.org/10.1148/radiol.220606>), speed of sound (<https://doi.org/10.1016/j.ultrasmedbio.2023.06.021>), and several envelope statistics parameters, are not included in the comparison. Also, the rationale for including only the quantitative USI parameters (attenuation coefficient and the acoustic structure quantification) and excluding the other parameters was not provided.

Reply:

As noted, the primary objective of this study was to explore the complementary roles of UFD and quantitative USI, rather than to provide a comprehensive comparison across all quantitative parameters. For this reason, we consider the use of ATI and ASQ, two widely adopted tools for diagnosing SLD, to be sufficient. Both techniques are well-established, supported by several clinical research reports, and implemented in commercially available US systems⁴⁻⁷.

While other quantitative parameters, such as the backscatter coefficient, speed of sound, and envelope statistics (e.g., Nakagami or normalized local variance imaging) are indeed valuable⁸⁻¹⁰, they were excluded due to methodological and practical considerations. Specifically, backscatter estimation requires precise calibration with reference phantoms, and speed of sound estimation requires specialized acquisition sequences. Further, obtaining accurate estimates using these two techniques necessitates access to raw channel data, which is generally unavailable in clinical ultrasound systems and, when combined with the aforementioned requirements, substantially increases computational complexity.

From the perspective of research efficiency, incorporating all techniques would have imposed a significant additional burden relative to the expected benefit. Therefore, we prioritized ATI and ASQ, which are more practical and clinically established. Additionally, ASQ reflects tissue microstructural properties by analyzing the statistical distribution of echoes, which indirectly captures tissue scattering aspects resembling the information obtained through backscatter coefficient, while being more efficient to implement.

For envelope statistics in the context of SLD diagnosis, additional methods such as Nakagami and normalized local variance imaging are available alongside ASQ. Although each method differs in its statistical model and mathematical formulation, they all share the underlying principle of leveraging the statistical distribution of speckle patterns. Based on prior research validating the diagnostic performance of ASQ, we adopted the method in this study.

*The **Discussion section on Pages 21–22** has been updated based on the above content to clarify the respective roles and purposes of UFD and quantitative USI:*

“The primary objective of this study was to explore the association between SLD progression, reflected by hepatic fat accumulation, and vascular alterations across multiple scales, encompassing both microvascular and macrovascular domains.

Accordingly, the study focused on investigating the complementary roles of UFD and quantitative USI, rather than conducting a direct comparison between them or aiming for precise hepatic fat quantification. In this context, quantitative USI was intended to serve as a complementary modality to UFD, enabling non-invasive confirmation of hepatic fat accumulation during SLD monitoring without reliance on histological analysis.”

“Among various quantitative USI techniques, two widely utilized tools for SLD diagnosis, ATI and ASQ, were employed in this study. Both techniques are well-established, supported by several clinical research, and implemented in commercially available US systems⁴⁻⁷. While other quantitative indices such as the backscatter coefficient, speed of sound, and envelope statistics are indeed valuable⁸⁻¹⁰, they were excluded due to methodological and practical considerations. Specifically, backscatter coefficient estimation requires precise calibration with reference phantoms, and speed of sound estimation requires specialized acquisition sequences. Furthermore, obtaining accurate estimates using these two techniques necessitates access to raw channel data, which is generally unavailable in clinical ultrasound systems and, when combined with the aforementioned requirements, substantially increases computational complexity. Other envelope statistics such as Nakagami and normalized local variance imaging differ from ASQ in their statistical models and mathematical formulations, yet share the fundamental principle of characterizing the statistical distribution of echo amplitude patterns. From the perspective of research efficiency, including all techniques would have imposed substantial additional burden with limited incremental benefit.”

2) The attenuation coefficient algorithm implemented in this study does not represent state of the art, which could lead to an underestimate of the performance of attenuation coefficient. The authors are encouraged to use state-of-the-art attenuation algorithms (e.g., <https://doi.org/10.1109/TUFFC.2017.2719962>, <https://doi.org/10.1109/TUFFC.2019.2903010>, <https://doi.org/10.1109/TUFFC.2022.3218920>) for a fairer comparison.

Reply:

Thank you for your valuable suggestion regarding the use of state-of-the-art attenuation coefficient estimation methods. We have carefully reviewed the recommended papers and updated our attenuation coefficient estimation approach by adopting the system-independent and phantom-free methodologies^{11, 12}. In the original manuscript, attenuation was estimated using a spectral shift-based method that tracked depth-dependent centroid frequency shifts. In the revised manuscript, we implemented a spectral normalization-based approach incorporating reference depth selection, frequency range constraints, and robust least-squares fitting to mitigate the influence of outliers. This updated implementation enables more system-independent and reliable estimation of the attenuation coefficient. The coefficient is now quantified in physical units (dB/cm/MHz), rather than relative values, reflecting methodological advances.

The new method yielded more stable trends in the revised SLD progression and recovery monitoring results (Fig. R1a,b,c). It also showed improved correlation and better distinction of grade 1, although with a slight reduction in grade 2 differentiation (Fig. R1d,e).

Fig. R1. Original and revised EAC components corresponding to (a) principle, (b) SLD progression monitoring, (c) SLD recovery monitoring, (d) correlation analysis, and (e) ROC curve analyses.

The Results (Figs. 1d, 3d, 4d, 5a, and Supplementary Fig. 9) and Methods sections have been updated accordingly.

Results: We have added the following sentences on Page 4:

“Within the region of interest (ROI), the power spectra along the depth windows were extracted, and the estimated attenuation coefficient (EAC) was derived from the linear regression of the natural logarithm of the normalized frequency power ratio.”

Methods: We have added the following sentences on Pages 27–28:

“We implemented the ATI function based on spectral normalization at different frequencies to analyze the attenuation from the selected ROI^{11, 12}. An area of the ROI measuring approximately 8 mm × 4 mm (width × height) was selected. The window size along the depth direction for the STFT was set to ten wavelengths, with an overlap of 97.5%. The STFT was applied along the depth axis for each column. The power spectra from multiple windows at the same depth was then averaged. Then, for each depth z_k , the ratio between the power spectrum values at two adjacent frequency bins (f_{i-1} , f_i) was calculated:

$$RS(f_i, z_k) = \frac{S(f_i, z_k)}{S(f_{i-1}, z_k)}, \quad (5)$$

where RS is the ratio of the power spectrum values, f_i is a frequency, z_k is a given depth, and S is the power spectrum. The normalized ratio of the power spectrum values at reference depth z_r was calculated as

$$RS_{nor}(f_i, z_k, z_r) = \frac{RS(f_i, z_k)}{RS(f_i, z_r)}, \quad (6)$$

where RS_{nor} is the normalized ratio of the power spectrum values and z_r is a reference depth. Because the attenuation is known to follow the equation

$$\ln[RS_{nor}(f_i, z_k, z_r)] = -4\alpha(f_i - f_{i-1})(z_k - z_r), \quad (7)$$

where α is the attenuation coefficient, α was estimated using the least squares method model implemented in the MATLAB function 'fitlm' with the 'RobustOpts' option. A consecutive frequency range was selected in which the estimated attenuation coefficient remained within $\pm 25\%$ of the value estimated at the frequency exhibiting the overall maximal power spectrum. A rectangular ROI was carefully selected to include only the area inside the LLL, and a total of five reliable US frames, covering large areas of the hepatic parenchyma, were selected to adequately represent tissue attenuation. The median value from the five calculated EACs was used as the representative EAC.”

3) A wide body of literature is available on artificial intelligence (AI)-based liver fat quantification using quantitative USI, which often yields highly accurate results. The superiority of UFD cannot be adequately supported without comparison with state-of-the-art AI-based quantitative USI.

Reply:

As mentioned earlier, our primary objective is to monitor the progression of SLD, reflected by fat accumulation, rather than to quantify hepatic fat content, per se, using UFD. In this context, quantitative USI techniques are not intended to compete with UFD, but rather to complement it by enabling non-invasive confirmation of fat accumulation without the need for histological analysis during SLD monitoring. Additionally, this work highlights the potential for integrating existing quantitative USI with UFD in multiparametric analyses. UFD offers vascular insights distinct from those of existing tissue-based quantitative USI, particularly in the context of SLD progression. Therefore, we believe that UFD can synergistically improve AI-based liver fat quantification using quantitative USI.

*We are concerned that the section of the manuscript suggesting that UFD has superior diagnostic performance to quantitative USI may have led to misunderstanding. Hence these statements have been revised or removed. Additionally, we have expanded the relevant sections of the **Discussion on Page 22** to more carefully address the context of the comparison between UFD and quantitative USI:*

“Our findings suggest that integrating UFD with quantitative USI parameters enhances the accuracy of predicting SLD development. Unlike conventional quantitative USI techniques that reflect tissue acoustic properties, UFD provides complementary information on hepatic vascularity. Recent studies have explored the use of deep learning models to accurately quantify hepatic fat by integrating multiple quantitative USI indices^{13, 14}. Vascular indices derived from UFD may offer valuable complementary features for AI-based fat quantification, providing information distinct from that captured by conventional quantitative USI indices.”

[Significance]

4) The study is significant in that it generates new knowledge that UFD-derived parameters could be highly correlated with liver fat. However, the significance in this aspect is slightly compromised by the fact that several studies have demonstrated the usefulness of conventional Doppler for SLD diagnosis and monitoring (<https://doi.org/10.1016/j.aohep.2020.09.008>). If conventional Doppler is useful, it is expected that UFD could be better than conventional Doppler. Prior studies have also reported that severity of fatty liver is associated with liver vascular injuries (<https://doi.org/10.1016/j.aohep.2020.09.008>), although the current study is more comprehensive.

Reply:

Thank you for highlighting the significant aspects of our manuscript. In conclusion, UFD can offer greater utility than conventional Doppler. While conventional Doppler has been utilized for SLD diagnosis, its relatively limited utility is evident in the paper you cited, where Doppler indices demonstrated poorer performance than B-mode ultrasound-based right liver lobe diameter measurements and indices derived from blood and systemic parameters.

This limitation likely arises from the fact that conventional Doppler US primarily detects macro-level changes in hepatic perfusion, such as portal hypertension and decreased hepatic artery resistivity, which are typically associated with large hepatic vessels including the portal vein, hepatic artery, and hepatic vein¹⁵⁻¹⁷. Vascular changes in SLD originate primarily in the hepatic microvasculature, such as the sinusoids, and the ability to detect these early alterations may enlarge the temporal horizon for SLD diagnosis.

In this context, UFD's visualization of the hepatic vascular system at both micro- and macro-vascular scales provides a more informed assessment of vascular alterations associated with SLD. In addition, our 3D UFD offers enhanced morphological insights by visualizing and quantifying three-dimensional vascular structures, providing richer information than the cross-sectional two-dimensional images of conventional Doppler US. Whereas conventional Doppler US is useful for confirming the presence of steatotic liver, this study longitudinally tracked its progression and demonstrated, for the first time, that the disease's advance is associated with integrated vascular changes involving both macro- and microvascular structures. These features position UFD as a more comprehensive and valuable tool than conventional Doppler US and suggest its potential as a practical modality for the diagnosis and monitoring of SLD.

*Based on the above, the **Discussion section on Pages 19–20** has been strengthened to further support the significance of our research findings.*

“Unlike conventional Doppler US, which is limited to observing phenomena such as portal hypertension and decreased hepatic artery resistivity in large hepatic vessels, UFD can visualize and quantify the comprehensive hepatic vascular system in 3D, encompassing both microvascular and macrovascular structures. This ability enables the detection of vascular alterations that emerge in the early stages of SLD, which may enhance the diagnostic utility. Notably, our 3D UFD approach, with enhanced morphological assessment, demonstrated for the first time that SLD progression is associated with integrated vascular alterations.”

5) Technical novelty of the manuscript seems limited, as UFD and quantitative USI are well-established methods and have previously been used in 3D as well.

Reply:

We acknowledge that the technical novelty of our manuscript may appear limited due

to the combination of well-established methods. However, this study is the first to longitudinally demonstrate the association between SLD and the microvasculature. To ensure high reproducibility across repeated measurements, we carefully structured the acquisition and analysis framework of the 3D UFD system and implemented clinically viable techniques (ATI and ASQ) for multiparametric analysis. In other words, we prioritized imaging robustness and developed assessment techniques to support this goal. The innovative significance of our 3D liver UFD, the first of its kind, is further described below.

The technical novelties in this manuscript include (1) real-time US-based respiratory gating and (2) a stable vascular modeling framework, both of which contribute to high measurement reliability. First, we developed real-time, US-based respiratory gating to enable efficient 3D ultrafast US acquisition. By applying this technique, 3D volume scanning, typically time-consuming and data-intensive, was optimized through the exclusion of respiratory motion and the acquisition of US data exclusively during cardiac cycle intervals; this approach minimized both the delay of imaging acquisition and the size of the stored volumetric data. Second, we established a stable vascular modeling framework to ensure consistent and precise volumetric image processing and quantification, achieved through the strategic application of various techniques at each relevant stage. Debiasing to eliminate system-specific effects, normalization to minimize inter-individual variability, non-local means filtering for noise reduction, histogram equalization to enhance contrast, and Jerman vesselness filtering to improve vascular connectivity were all appropriately implemented.

These technical developments enabled a high level of robustness by effectively minimizing variability across numerous imaging procedures. As a demonstration, we examined the inter-subject reproducibility among rats in the steady state at week 0 ($n = 17$: i.e., $n = 12$ from SLD progression and recovery monitoring schemes, and $n = 5$ from the SLD validation scheme), which can be considered to reflect equivalent experimental conditions. The UFD indices demonstrated overall good reproducibility (Fig. R2). Low coefficients of variation (CVs) were observed for wnVVO and wnFMBV, and very low CV for SOAM, indicating excellent measurement reliability.

Fig. R2. Inter-subject reproducibility of UFD indices in the steady-state rats ($n = 17$, at week 0). CV, coefficient of variation.

Moreover, to the best of our knowledge, this is the first study to achieve precise

longitudinal 3D vascular imaging of the liver through repeated in vivo imaging enabled by these techniques. In fact, we needed a total of at least 133 imaging procedures to execute all three experimental schemes in this study: SLD progression monitoring (Supplementary Fig. 5a, a minimum of 72 imaging procedures), SLD validation (Supplementary Fig. 5b, a minimum of 25 imaging procedures), and SLD recovery monitoring (Supplementary Fig. 10, a minimum of 36 imaging procedures).

*The above content has been newly incorporated into the **Discussion section on Page 19**, and a corresponding figure has been added to **Supplementary Figure 17**, as we believe this material is essential to demonstrate the technical strengths of the study.*

“To the best of our knowledge, this is the first study to observe and quantify the overall vascular alterations associated with SLD in such a detailed and longitudinal fashion. Several technical novelties introduced to enhance imaging robustness enabled successful long-term volumetric scan observations. First, volume scanning was optimized using real-time ultrasound-based respiratory gating, which enabled US acquisition exclusively during cardiac cycle intervals, facilitating efficient 3D Ultrafast US acquisition. Second, by strategically applying various techniques (debiasing, intensity normalization, non-local means filtering, histogram equalization, and Jerman filtering) at each relevant stage, a stable vascular modeling framework was established, enabling consistent and accurate volumetric image processing and quantification. Indeed, our UFD indices demonstrated excellent inter-individual reproducibility and inter-system agreement, reflecting overall high reproducibility (Supplementary Figs. 17 and 18). This robustness enabled the stable execution of the approximately 133 imaging procedures required for the study, including SLD progression monitoring, SLD validation, and SLD recovery monitoring, with a minimum of 72, 25, and 36 procedures, respectively (Supplementary Figs. 5 and 10).”

6) The potential clinical impact of the study is uncertain given the potential challenges associated with performing UFD scans in humans, as the authors acknowledged. The authors mentioned several challenges, including the need for accelerated data processing (the current total imaging time is 10-12 minutes) and the challenges in obtaining 3D images from the human abdomen. While the challenge in imaging time is easier to address, the challenge in 3D imaging is more difficult to overcome due to the limited acoustic window size for liver imaging, particularly when an intercostal approach is used for liver imaging. Further, there are several fundamental challenges that the authors did not discuss that may significantly impact clinical translation. The impressive UFD images presented in the manuscript were acquired at a center frequency of 18 MHz, with a pulse repetition frequency of 20 kHz and an imaging depth of a few millimeters. For clinical human liver imaging, however, the center frequency is typically around 3 MHz. The significant reduction in the ultrasound frequency will lead to not only significantly worse image resolution, but also significantly lower sensitivity to blood flow, both of which lead to challenges in extrapolating the performance of UFD from rats to humans. In addition, the drastically increased imaging depth required in human liver imaging will lead to a reduction of pulse repetition frequency, which will adversely affect UFD image quality.

Reply:

We fully acknowledge the reviewer's concern that the potential clinical impact of our study is uncertain due to potential technical challenges in performing UFD scans in humans. However, we believe that these challenges can be overcome through the following measures, and we remain confident in the potential of UFD for effective application in future clinical studies. Naturally, substantial advances in signal processing and hardware will be required to implement these approaches.

In line with the reviewer's comment, imaging time can be easily reduced through accelerated data processing facilitated by GPU-based computation. For 3D imaging, the use of matrix probes, commonly employed in recent brain imaging, and fan-shaped tilt scanning (e.g., the Wobbler mechanism), widely adopted in conventional volumetric abdominal US, may enable the acquisition of reliable volumetric images even within a limited acoustic window¹⁸⁻²¹.

*As the reviewer pointed out, the limited resolution associated with low-frequency US probes, typically operating at a center frequency of approximately 3 MHz for abdominal imaging, and the reduction in pulse repetition frequency due to increased imaging depth are critical concerns that may significantly affect the practical implementation of UFD. However, given that the scale of blood vessels in the human vascular system is approximately ten times larger than in the rat vascular system, a center frequency of 3 MHz is expected to be sufficient for visualizing the human microvasculature^{22, 23}. Additionally, as the resting heart rate in humans (60–100 bpm) is substantially lower than that in rats (250–400 bpm), it is feasible to achieve a sufficient ensemble length even with a lower pulse repetition frequency than in rat imaging, thereby preserving vascular sensitivity. In practice, we are currently preparing an extended study on human liver imaging, utilizing the same Vantage 256 system along with GE C1-6-D probe to acquire high-quality human liver images. At a frame rate of 500 Hz, images with an ensemble length of 400 frames (0.8 seconds) have been acquired via the intercostal window (**Fig. R3**). As shown in the figure, the human hepatic vasculature is visualized in sufficient detail across an extensive region.*

Fig. R3. Human liver UFD image using a GE C1-6-D probe.

High time-bandwidth product pulses, such as coded excitation, which has already been implemented in various ultrasound applications, can further enhance human liver imaging. To address the frequency and imaging depth trade-offs, pulse compression techniques may improve the signal-to-noise ratio at greater imaging depths, thereby enabling the use of linear transducers with higher center frequencies. These improvements could further enhance the potential applications of UFD in clinical human liver imaging.

*The **Discussion** section on **Pages 23–24** has been reinforced to clearly highlight the potential of human liver UFD, based on the content outlined above. **Figure R3** has been included as **Supplementary Figure 19**.*

*“Extending our research to extensive clinical studies could address these limitations. In practice, we are currently preparing an extended study on human liver imaging, utilizing the same Vantage 256 system along with a GE C1-6-D probe to acquire high-quality human liver images. At a frame rate of 500 Hz, images with an ensemble length of 400 frames (0.8 seconds) have been acquired via the intercostal window (**Supplementary Fig. 19**). As shown in the figure, the human hepatic vasculature is visualized in sufficient detail across an extensive region.*

However, technical challenges must still be carefully considered and overcome. For effective monitoring and medical intervention in near real-time, the UFD procedure requires accelerated data processing through GPU-based computation. Further, although our rat-model study was conducted in 3D, obtaining 3D images from the human abdomen is challenging. Using matrix probes or fan-shaped tilt scanning (e.g., the Wobbler mechanism) may enable the acquisition of reliable volumetric images even within a limited acoustic window¹⁸⁻²¹. Although 2D image acquisition may be

considered, both 2D and 3D imaging would require motion compensation to account for respiration^{24, 25}. In human abdominal imaging, low-frequency US probes with a central frequency of approximately 3 MHz are typically used, and the imaging depth will be substantially greater than in rat imaging. This difference can reduce the spatial resolution and pulse repetition frequency (PRF), potentially compromising UFD image quality. However, given the differences in vascular scale and heart rate between humans and rats^{22, 23}, a reduced PRF may still provide a sufficient ensemble length to compensate for potential degradation in image quality. The human liver, located deep within the abdomen, can be thick enough to impede US beam transmission, necessitating techniques such as pulse compression in UFD^{26, 27}. High time-bandwidth product pulses, such as coded excitation, which has already been implemented in various US applications, can enhance the signal-to-noise ratio at greater imaging depths by mitigating the trade-off between frequency and penetration, thereby enabling the use of linear transducers with higher center frequencies. Additionally, given the complexity of the liver vasculature, color Doppler techniques, which can specify the blood flow direction and velocity, would be highly functional. Ultimately, by building on technical advances, integrating UFD into clinical practice could expand its applications to surgical procedures such as liver transplantation, in addition to diagnosis and monitoring of various diseases, such as hepatitis B, cirrhosis, and hepatocellular carcinoma.”

7) Existing quantitative USI methods have already shown good accuracy in liver fat quantification. The added value of UFD is unclear if the superiority of UFD is not adequately supported. More importantly, primary challenges with some of the existing USI methods are their lack of reproducibility, as shown in <https://doi.org/10.1148/radiol.240162>. The reproducibility of UFD was not addressed in the manuscript, which casts further doubts in its added value. UFD-derived parameters are dependent on numerous signal and image processing parameters such as the singular value decomposition cutoff values and vascular modeling processes. Therefore, it seems crucial to address the reproducibility of UFD-derived parameters in this study to demonstrate inter-operator and inter-system agreement.

Reply:

We very much appreciate your valuable suggestions on enhancing the added value of UFD, and your comment makes it clear that we have inadequately conveyed its performance strengths. Overall, our UFD approach demonstrates excellent reproducibility, supported by technical novelties aimed at enhancing imaging robustness (please see the response to Comment 5, above). Consistent with the high inter-subject reproducibility, inter-operator and inter-system agreement are also strong.

Our imaging system secures the position of the US probe via hardware configuration and performs scanning through mechanical translation, thereby limiting inter-operator variability to the determinations of the scanning position and range. Additionally, we enhanced the robustness of the process by positioning the rat in a fixed, supine position on the imaging bed. To further eliminate variability during the

quantitative evaluation stage, all signal and image processing parameters were fixed and kept constant. For example, a fixed cutoff was applied in SVD-based clutter filtering, and all parameters used for vascular modeling were consistently maintained. Consequently, inter-operator variability in our imaging system is considered minimal.

Regarding inter-system agreement, verification on other US systems presents practical challenges, as the imaging platform used in this study is a programmable, research-grade US system. Commercial US systems offer restricted support for custom code integration, and other research-grade US systems are only partially programmable; moreover, neither system is available to us. Instead, we own the Vantage NXT platform, which features specifications comparable to the 256-channel system (Vantage 256, Verasonics, USA) used in this study. Therefore, we conducted additional inter-system agreement verification experiments using both platforms on subjects from the normal and SLD state at week 4 ($n = 12$, $n = 6$ for each group).

Using both the Vantage High and NXT platforms, UFD successfully acquired volumetric images that were nearly identical in normal and SLD at week 4 (**Fig. R4a**). To quantitatively assess inter-system agreement, we employed the intraclass correlation coefficient (ICC), Pearson's correlation coefficient, and Bland-Altman analysis (**Fig. R4b**). Although this was an exploratory-level validation of inter-system agreement, all ICC values and Pearson's correlation coefficients exceeded 0.8, and all data points fell within the limits of agreement, indicating a high level of agreement reliability.

As noted in the response to Comment 5, **Figure R4** has been included as **Supplementary Figure 18** to further illustrate the technical strengths of this study.

Fig. R4. Results of inter-system agreement experiments. **a**, Volumetric PD images acquired using Vantage High and NXT systems for Normal and SLD rats at week 4. **b**, Scatter and Bland-Altman plots comparing the UFD quantification results, including the intraclass correlation coefficient and Pearson's coefficient. SLD, steatotic liver disease; wnVVO, weight-normalized vessel volume occupancy; wnFMBV, weight-normalized fractional moving blood volume; VBD, vessel bifurcation density; SOAM, sum of angles metric; ICC, intraclass correlation coefficient; PC, Pearson's correlation coefficient; SD, standard deviation, and diff, difference.

[Data and methodology]

Overall the animal study design and the UFD data acquisition and processing are solid, and the presentation of the manuscript is reasonable. However, a few issues and potential flaws are noted, and discussed as follows.

8) The authors developed a clever method for automatic ensemble selection during UFD processing (Supplemental Figure 1b), which is excellent. However, it appears that the cardiac cycle was not considered in their data analysis. Blood flow is sensitive to the cardiac cycle, which indicates that UFD-derived parameters may be dependent on the cardiac cycle. Ignoring the cardiac cycle may lead to increased variability in the UFD-derived parameters.

Reply:

Once again, thank you for your insightful comments. As you noted, blood flow is indeed sensitive to the cardiac cycle. However, our UFD-derived parameters were acquired under a relatively consistent cardiac phase, specifically the diastole. The results of simultaneous UFD–electrocardiogram (ECG) measurements, acquired through additional experiments, are presented below.

*We established automatic ensemble selection criteria with the aim of minimizing the cardiac cycle dependence of UFD-derived parameters. When accounting for the time lag, the peak of the B-mode interframe error corresponds to the R peak in the ECG signal (**Fig. R5a**). The ensemble length was defined as the interval following the T wave and preceding the subsequent QRS complex. This approach was intended to capture diastole flow following ventricular systole, which was particularly important given the need to visualize microvascular flow in our imaging (**Fig. R5b**).*

Fig. R5. Correlation between the cardiac cycle and B-mode interframe error. a, ECG signal and corresponding interframe error over time. **b,** PD image depicting steady-state blood flow.

*Your point regarding potential variations in PD plots across cardiac cycles in the normal and SLD groups offers valuable additional insights. Accordingly, we have incorporated this point into the **Methods section on Page 26**:*

“After the planewave US acquisition, the most stable 150 frames corresponding to a relatively consistent cardiac phase were selected as an ensemble from the 600 US IQ frames obtained at each location (Supplementary Fig. 1c-e and Supplementary Video 2).”

“To capture diastole flow after ventricular systole, 150 sequential frames avoiding peaks caused by heartbeats were selected.”

9) The UFD data acquisition workflow could be optimized to allow faster imaging. First example, instead of waiting until all acquired data have been beamformed to move to the next scan location, the authors could have acquired all the data first and performed beamformed through post processing, which would significantly reduce the data acquisition time.

Reply:

Thank you for your valuable feedback. However, the purpose of performing beamforming on all acquired data during the imaging procedure at each scan location is to enable immediate visualization of imaging results through offline post-processing once the online imaging process—including data saving—is completed. The liver may be affected by surrounding organs such as the heart, intestines, and stomach, and organ motion can cause certain frames to become out-of-plane, necessitating re-acquisition. Therefore, this process is essential for verifying the accuracy of the imaging results. Ultimately, because beamformed data needed to be generated, the use of channel data would have significantly increased the time required for saving and loading the acquired data: In practice, beamformed data reconstruction requires approximately 6.2 seconds, and storage takes about 0.3 seconds at a single scan position, whereas storing channel data takes approximately 1.1 seconds. Although the duration of the online imaging process could be reduced, the use of channel data would eventually increase the total time required for saving and loading, due to the need for beamforming reconstruction during post-processing.

Additionally, in terms of storage requirements, channel data (~2.6GB) occupies approximately 3.7-times more space than beamformed data (~734MB). Considering the number of scanning positions (~100), this results in a difference of approximately 190GB per imaging procedure. Given our experimental design, which involves over 100 imaging procedures, this imposes substantial data storage requirements, presenting a considerable practical challenge.

10) There appears to be a flaw in how the “all-around US score” was derived. The authors “tested all possible combinations of the six US indices, a total of 63 cases (Supplementary Table. 3)”, which should be viewed as a feature selection process in machine learning terminology. This feature selection process should be performed using a separate dataset, which was not the case in this manuscript. Even though the authors performed cross-validation to evaluate the performance and check overfitting, the cross-validation process performed by

the authors did not involve feature selection, and therefore potential bias in feature selection could still lead to an over-estimation of the performance.

Reply:

Thank you for highlighting the limitations of our all-around US score. We fully agree with your comments. To enhance the model, we have revised the cross-validation process and restructured the overall framework to enable feature selection using separate datasets for testing and validation. Using a nested Monte Carlo cross-validation framework^{28, 29}, we performed 100 iterations of the entire process, consisting of both outer and inner loops.

In the outer loop, the entire dataset ($n = 37$) was divided into a training set (80%) and a test set (20%). Then, 5-fold cross-validation was performed within the inner loop, using only the training set defined in the outer loop. The final model, trained using the averaged weights and biases from each fold, was then evaluated on the test set from the outer loop.

*Feature selection was carried out exclusively based on the 5-fold cross-validation performance within the inner loop, where the average performance of all possible combinations derived from the six US indices was evaluated (**Fig. R6a**). Through this process, the combination of five indices (wnVVO, wnFMBV, VBD, SOAM, and EAC) was identified as the optimal subset. Additionally, to evaluate the model's generalization performance, we calculated the average test set performance across all possible combinations in each iteration of the nested Monte Carlo cross-validation (**Fig. R6b**). Numerous combinations yielded Pearson correlation coefficients exceeding 0.9. Moreover, as the number of combined indices increased, performance became more stable, indicating robust generalization capability.*

Fig. R6. Performance of nested Monte Carlo 5-fold cross-validation based on combinations of indices. a, Validation fold performance used to select the optimal feature combination. **b,** Test set performance for evaluating generalization capability. Data are presented as mean \pm standard deviation. A list of combinations is provided in **Supplementary Table 6**.

Since feature selection and generalization performance were evaluated using the nested Monte Carlo cross-validation SVR, we further assessed the discriminative capacity of the selected combination by applying a single SVR model to the same dataset. **Fig. 5b** presents the results based solely on changes in the index combination while maintaining the same structure, and **Supplementary Fig. 12** has been updated accordingly. The **Results and Methods sections** have been revised to reflect the updated methodology.

Results: We have added the following sentences on Page 17.

“To mitigate overfitting, we employed a nested Monte Carlo cross-validation approach to select the optimal combination using a separate dataset. The performance of SVR models is evaluated by PC, and this process identified a combination of five indices (wnVVO, wnFMBV, VBD, SOAM, and EAC) as the optimal subset (Supplementary Fig. 16a).

The combination of five indices (wnVVO, wnFMBV, VBD, SOAM, and EAC) exhibits a PC of 0.96 and a SC of 0.95 and demonstrates an average balanced accuracy of 92% in all classifications, which exceeds that of all US indices (Fig. 5b). Although the performance improvements within the SVR models are modest due to the excellent individual capabilities of wnVVO and wnFMBV, considering the variability of the indices, this multi-index interactive all-around US score would provide more consistent performance across various measurement conditions.

Additionally, to evaluate the model's generalization performance, we calculated the average test set performance across all possible combinations in each iteration of the nested Monte Carlo cross-validation (Supplementary Fig. 16b). Numerous combinations yielded PC values exceeding 0.9. Moreover, as the number of combined indices increased, performance became more stable, indicating robust generalization. These findings confirm the robustness of our approach and demonstrate that UFD indices can be harmonized with conventional US indices (See Methods).”

Methods: we have added the following sentences on Pages 33–34.

“Nested Monte Carlo cross-validation was employed to identify the optimal feature subset and assess the model's generalization performance while minimizing the risk of overfitting^{28, 29}. In the outer loop, all 37 data were divided into a training set (80%) and a test set (20%). Subsequently, 5-fold cross-validation was conducted within the inner loop, using only the training set. The performance results obtained using four folds for training and one fold for validation were used exclusively to select the optimal combination. Thereafter, the model, trained using the averaged weights and biases from the 5-fold cross-validation, was evaluated on the test set in the outer loop, and this performance was used to assess generalization. This process was repeated 100 times; namely, the whole loop was executed 100 times. Finally, a single SVR model was trained using the selected optimal feature combination to enable comparison with the original six US indices.”

II) Figure 3-g showed a moderate correlation (Pearson correlation coefficient: -0.71; Spearman's correlation coefficient, -0.63) between CD31 density (direct quantification of microvascular damage) and fat percentage, and the correlation was weaker than those between the UFD-derived parameters and fat percentage (please see “Hemodynamics” and “Vascular morphology” plots in Figure 5-a). Given that the CD31 density is a direct quantification of microvascular damage, and UFD-derived parameters are only surrogate measures of

microvascular damage, can the authors provide insight into why the UFD-derived parameters showed a higher correlation with fat percentage than CD31 density?

Reply:

We would like to emphasize that UFD-derived parameters reflect not merely the presence of blood vessels, but also the in-vivo blood flow within the vessels of the integrated vascular system. In other words, UFD-derived parameters, rather than functioning solely as surrogate measures of microvascular damage, represent an effective integrated vascularity index from the perspective of vascular dynamics. UFD-derived parameters may provide a more comprehensive reflection of SLD progression than CD31 density, which could account for their higher correlation coefficients with fat percentage.

*We believe this explanation will improve the clarity of the manuscript for readers and have therefore incorporated it into the **Discussion section on Page 21.***

“We would like to emphasize that UFD indices reflect not merely the presence of blood vessels, but the in-vivo blood flow within the integrated vascular system, thereby representing an effective vascularity index from the perspective of vascular dynamics. UFD indices may provide a more comprehensive reflection of SLD progression than CD31 density, which could account for their higher correlation coefficients with fat percentage.”

12) Lines 392-395: Can the authors clarify how a pulse repetition frequency of 20 kHz combined with 11-angle planewave pulses resulted in a B-mode frame rate of 1000 Hz? The math did not seem right, or I might be missing something here.

Reply:

*We apologize for the confusion. In 11-angle US transmission, each angle's pulse is transmitted at a 50 μ s interval, yielding a planewave pulse repetition frequency of 20 KHz (**Fig. R7**). However, a brief resting period (450 μ s) is inserted between consecutive 11-angle transmissions to ensure probe stability, resulting in a B-mode frame rate of 1000 Hz.*

Fig. R7. 11-angle planewave US transmission scheme.

*For clarification, this figure has been included in **Supplementary Figure 1** and the main text.*

Results: *A reference to the figure has been added on Page 4.*

*“At a single imaging location on the rat’s abdomen, respiration was traced in real-time via B-mode interframe errors, automatically triggering an 11-angle planewave US transmission (-10°–10°) to construct an in-phase/quadrature (IQ) frame (**Fig. 1b**, **Supplementary Figs. 1, 2a,b**, and **Supplementary Video 1**).”*

13) Line 396: The step size was 0.2 mm in the elevational direction of the transducer, which was 1/8 of the transducer’s elevational width of 1.6 mm (<https://verasonics.com/high-frequency-transducers/>). Can the authors justify the use of such a small step size?

Reply:

*A dense 3D volumetric scan step size was selected to enhance visualization of vascular network connectivity and to enable more precise implementation of vascular morphological indices. Although the transducer specifications indicate an elevational width of 1.6 mm, the actual width of the US beam differs from this nominal value. According to the Rayleigh approximation³⁰, a mathematical model for measuring the elevational beam width of a transducer, the elevational beam width is approximately equal to $(c/f)*F/H$, where c is speed of sound, f is the US frequency, F is the elevation focus, and H is the elevation width. In our case, $c = 1500$ m/s (the sound speed at which macrovascular structures were optimally visualized in SLD weeks 3–4 B-mode liver imaging), $f = 15.625$ MHz (carrier frequency), $F = 8$ mm, and $H = 1.6$ mm. Therefore, the theoretically estimated elevational beam width is approximately 0.48 mm, which is consistent with the value reported in the literature³¹. Accordingly, considering spatial sampling based on the Nyquist sampling theorem, a minimum*

elevational step size of 0.24 mm (half the beam width) is required to precisely reconstruct signals at spatial locations. Taking into account the pixel size (0.05 mm) in our 2D imaging setup and the need for acquisition efficiency, an elevational step size of 0.2 mm was selected.

*This content has been added to the **Methods section on Page 25** and newly organized as **Supplementary Note 1** to provide a detailed explanation of the technical background underlying our research.*

*“Based on the transducer specifications, a dense scanning step size of 0.2 mm was set to better visualize vascular network connectivity and enable more precise computation of vascular morphological indices (**Supplementary Note 1**).”*

14) Line 448: “the window size along the depth direction and the overlap length were optimized experimentally” – can the authors clarify the optimization process? This detail is important to ensure transparency.

Reply:

Thank you for your comment regarding the clarification of the optimization process. As mentioned, we have updated the attenuation coefficient estimation method. Accordingly, the sentence in Line 448 has also been revised. In the updated method, short-time Fourier transform (STFT) is performed along the axial direction by segmenting the selected region of interest (ROI) into units with a length of 10 wavelengths. An axial overlap length of 97.5% between adjacent segments is used.

*We have revised the Methods section to clearly describe these parameters in accordance with your suggestion to ensure greater transparency. Please refer to the updated **Methods section on Pages 27–28** for further details. This content is also addressed in the response to Comment 2; please refer to that section for details.*

15) Can the authors clarify why support vector regression (SVR), as opposed to other methods, was used to combine the ultrasound indices?

Reply:

Given the limited number of samples ($n = 37$) and features ($n = 6$) in our dataset, we employed support vector regression (SVR). In contrast to conventional regression models, SVR is grounded in the principle of structural risk minimization, which aims to balance empirical error and model complexity³². SVR also leverages margin-based learning and constructs predictive models using only a subset of the training data—namely, the support vectors—which enhances robustness to overfitting and reduces sensitivity to noise. Owing to these characteristics, even with limited data, strong generalization performance can be expected with SVR.

*Based on the above, we have revised the description of SVR in **the Methods section on Page 33**.*

“SVR was adopted due to its robustness to overfitting and noise, even with limited data, and its strong generalization performance³².”

16) The sample size does not seem to be adequately justified. The sample size of 37 rats seems small compared to the types of statistical analyses performed in the study; however, it is beyond the scope of my expertise to scientifically assess the appropriateness of this sample size.

Reply:

We appreciate your insightful comments regarding the study design. With respect to the sample size, we kindly refer you to our response to the Editor’s comment provided at the beginning of this letter.

*Regarding the types of statistical analyses, we revised the methodology to more accurately evaluate the results of the SLD progression and recovery monitoring scheme. Student’s *t*-test was initially employed under the assumption of normal distribution and equal variance between groups. However, due to the small sample size ($n = 4$ per group), the normality assumption could not be reliably validated. Therefore, we updated our statistical approach to the two-sided Wilcoxon rank-sum test, a non-parametric method more suitable for small sample sizes.*

*These changes have been incorporated into the former **Supplementary Tables 1 and 2**. Currently, the Supplementary Information have been updated and renumbered, and the tables have been separated into **Supplementary Tables 2–4** to reflect the inclusion of additional LYVE1 staining, as recommended by Reviewer #2:*

Supplementary Table 2. Statistical analyses of the SLD progression monitoring experiments. Quantitative US indices were monitored over 8 weeks. Weight-normalized vessel volume occupancy, wnVVO; weight-normalized fractional moving blood volume, wnFMBV; vessel bifurcation density, VBD; sum of angles metric, SOAM; estimated attenuation coefficient, EAC; and focal disturbance ratio, FDR. Blood biomarkers such as alanine aminotransferase (ALT), aspartate transaminase (AST), and total bilirubin (TBIL) were tested. *p*-values are calculated using the two-sided Wilcoxon rank-sum test, and values ≤ 0.05 are highlighted in yellow.

US index	Group	Week 0	Week 1	Week 2	Week 3	Week 4	Week 5	Week 6	Week 7	Week 8
wnVVO [a.u.]	Normal	44.16 ± 1.28	47.73 ± 1.42	46.60 ± 0.26	45.07 ± 0.89	43.47 ± 1.98	45.34 ± 0.89	43.38 ± 1.91	46.25 ± 0.92	44.68 ± 1.15
	SLD	44.24 ± 1.29	38.01 ± 1.61	37.41 ± 0.55	28.66 ± 1.19	26.71 ± 2.69	26.83 ± 1.51	29.72 ± 0.31	27.68 ± 0.62	28.41 ± 0.63
p -value	Norm. - SLD	1.000	0.029*	0.029*	0.029*	0.029*	0.029*	0.029*	0.029*	0.029*
wnFMBV [a.u.]	Normal	29.64 ± 1.63	32.03 ± 0.84	30.93 ± 0.39	31.65 ± 0.89	29.30 ± 1.22	31.86 ± 0.25	31.10 ± 1.60	32.88 ± 0.54	31.07 ± 0.16
	SLD	29.32 ± 0.91	25.06 ± 1.18	24.43 ± 0.58	18.50 ± 0.62	17.54 ± 1.64	17.02 ± 1.05	19.59 ± 0.28	17.98 ± 0.45	19.05 ± 0.26
p -value	Norm. - SLD	0.686	0.029*	0.029*	0.029*	0.029*	0.029*	0.029*	0.029*	0.029*
VBD [mm ⁻³]	Normal	4.12 ± 0.37	4.73 ± 0.37	4.48 ± 0.23	3.82 ± 0.06	4.01 ± 0.33	3.79 ± 0.18	4.01 ± 0.19	4.11 ± 0.38	4.20 ± 0.21
	SLD	4.38 ± 0.36	3.28 ± 0.24	3.31 ± 0.15	2.22 ± 0.41	1.87 ± 0.43	1.61 ± 0.33	1.98 ± 0.15	2.18 ± 0.06	1.65 ± 0.28
p -value	Norm. - SLD	0.686	0.029*	0.029*	0.029*	0.029*	0.029*	0.029*	0.029*	0.029*
SOAM [-]	Normal	8.57 ± 0.03	8.60 ± 0.01	8.56 ± 0.03	8.62 ± 0.02	8.58 ± 0.03	8.60 ± 0.05	8.51 ± 0.03	8.55 ± 0.06	8.57 ± 0.03
	SLD	8.50 ± 0.02	8.67 ± 0.04	8.67 ± 0.03	8.69 ± 0.08	8.72 ± 0.06	8.86 ± 0.02	8.86 ± 0.03	8.77 ± 0.03	8.86 ± 0.06
p -value	Norm. - SLD	0.057	0.686	0.114	0.686	0.057	0.029*	0.029*	0.057	0.029*
EAC [a.u.]	Normal	0.82 ± 0.01	0.79 ± 0.03	0.81 ± 0.05	0.76 ± 0.07	0.66 ± 0.04	0.79 ± 0.05	0.82 ± 0.04	0.75 ± 0.03	0.77 ± 0.05
	SLD	0.80 ± 0.04	0.89 ± 0.02	0.86 ± 0.02	0.90 ± 0.05	0.95 ± 0.03	0.98 ± 0.03	1.09 ± 0.06	1.13 ± 0.05	1.27 ± 0.05
p -value	Norm. - SLD	0.886	0.057	1.000	0.200	0.029*	0.029*	0.029*	0.029*	0.029*
FDR [-]	Normal	0.63 ± 0.03	0.64 ± 0.03	0.62 ± 0.04	0.55 ± 0.04	0.61 ± 0.03	0.56 ± 0.04	0.58 ± 0.04	0.57 ± 0.04	0.63 ± 0.03
	SLD	0.72 ± 0.02	0.52 ± 0.01	0.45 ± 0.01	0.42 ± 0.03	0.43 ± 0.01	0.42 ± 0.02	0.45 ± 0.02	0.42 ± 0.01	0.40 ± 0.01
p -value	Norm. - SLD	0.114	0.029*	0.029*	0.114	0.029*	0.029*	0.057	0.029*	0.029*

Blood test	Group	Week 0	Week 1	Week 3	Week 5	Week 7	Week 8
ALT [U/L]	Normal	38.25 ± 2.02	41.50 ± 2.25	29.00 ± 3.72	16.25 ± 3.20	35.50 ± 1.94	33.00 ± 1.41
	SLD	41.25 ± 4.05	40.25 ± 9.94	133.75 ± 30.41	199.00 ± 49.49	124.25 ± 18.07	146.00 ± 40.51
p -value	Norm. - SLD	0.686	0.571	0.029*	0.029*	0.029*	0.029*
AST [U/L]	Normal	102.50 ± 16.32	73.25 ± 2.02	150.75 ± 50.19	60.00 ± 7.33	82.75 ± 15.98	88.75 ± 15.47
	SLD	79.00 ± 15.20	102.25 ± 23.69	184.00 ± 24.53	177.00 ± 25.06	172.50 ± 29.47	161.50 ± 24.02
p -value	Norm. - SLD	0.486	1.000	0.686	0.029*	0.343	0.057
TBIL [mg/dl]	Normal	0.20 ± 0.04	0.18 ± 0.03	0.20 ± 0.06	0.25 ± 0.03	0.38 ± 0.05	0.28 ± 0.05
	SLD	0.25 ± 0.03	0.25 ± 0.03	0.63 ± 0.05	0.65 ± 0.10	0.78 ± 0.13	0.88 ± 0.18
p -value	Norm. - SLD	0.657	0.286	0.029*	0.029*	0.171	0.029*

Supplementary Table 3. Histological quantification results of the SLD progression monitoring and validation experiments. Hepatic fat percentage was quantified from ORO staining, and microvascular damage was quantified by CD 31 and LYVE1 staining. *p*-values are calculated using the two-sided Wilcoxon rank-sum test, and values ≤ 0.05 are highlighted in yellow.

Histological quantification	Normal week 0	SLD weeks 1–2	SLD weeks 3–4	SLD week 8	Normal week 8
Fat [%]	1.42 ± 0.44	32.88 ± 4.75	47.69 ± 2.65	68.01 ± 4.94	0.81 ± 0.11
p -value (vs Normal week 0)		0.0007**	0.0007**	0.016*	0.413
p -value (vs SLD weeks 1-2)			0.064	0.002**	0.002**
p -value (vs SLD weeks 3-4)				0.004**	0.002**
p -value (vs SLD week 8)					0.029*

Histological quantification	Normal week 0	SLD weeks 1–2	SLD weeks 3–4	SLD week 8	Normal week 8
CD31 density [%]	36.77 ± 1.10	34.85 ± 1.11	26.28 ± 1.03	22.67 ± 1.40	34.12 ± 1.12
p -value (vs Normal week 0)		0.513	0.0007**	0.016*	0.286
p -value (vs SLD weeks 1-2)			0.0004**	0.002**	0.839
p -value (vs SLD weeks 3-4)				0.076	0.002**
p -value (vs SLD week 8)					0.029*

Histological quantification	Normal week 0	SLD week 2	SLD week 4	SLD week 8	Normal week 8
LYVE1 density [%]	43.68 ± 2.31	30.61 ± 1.60	28.16 ± 0.61	29.88 ± 3.00	42.62 ± 1.28
p -value (vs Normal week 0)		0.029*	0.029*	0.029*	0.488
p -value (vs SLD week 2)			0.2	0.886	0.029*
p -value (vs SLD week 4)				0.886	0.029*
p -value (vs SLD week 8)					0.029*

Supplementary Table 4. Statistical analyses of the SLD recovery monitoring experiments. Quantitative US indices were monitored over 8 weeks. Weight-normalized vessel volume occupancy, wnVVO; weight-normalized fractional moving blood volume, wnFMBV; vessel bifurcation density, VBD; sum of angles metric, SOAM; estimated attenuation coefficient, EAC; and focal disturbance ratio, FDR. Blood biomarkers such as alanine aminotransferase (ALT), aspartate transaminase (AST), and total bilirubin (TBIL) were tested. *p*-values are calculated using the two-sided Wilcoxon rank-sum test, and values ≤ 0.05 are highlighted in yellow.

US index	Week 0	Week 1	Week 2	Week 3	Week 4	Week 5	Week 6	Week 7	Week 8
wnVVO [a.u.]	45.19 ± 0.71	38.63 ± 0.69	37.58 ± 1.78	32.80 ± 2.01	32.37 ± 1.67	34.83 ± 1.47	38.51 ± 1.32	40.46 ± 1.38	41.64 ± 0.51
p -value (vs Norm.)	0.686	0.029*	0.029*	0.029*	0.029*	0.029*	0.114	0.057	0.057
p -value (vs SLD.)	0.686	1.000	0.886	0.200	0.343	0.029*	0.029*	0.029*	0.029*
wnFMBV [a.u.]	30.45 ± 1.19	25.42 ± 0.22	24.52 ± 1.05	21.71 ± 1.13	21.55 ± 0.89	23.96 ± 1.27	27.67 ± 0.83	27.83 ± 0.65	28.57 ± 0.34
p -value (vs Norm.)	0.686	0.029*	0.029*	0.029*	0.029*	0.029*	0.200	0.029*	0.029*
p -value (vs SLD.)	0.686	0.486	1.000	0.114	0.114	0.029*	0.029*	0.029*	0.029*
VBD [mm ⁻³]	4.49 ± 0.20	4.03 ± 0.22	3.39 ± 0.33	2.61 ± 0.36	2.28 ± 0.27	2.32 ± 0.27	2.85 ± 0.30	3.68 ± 0.36	3.59 ± 0.25
p -value (vs Norm.)	0.486	0.343	0.057	0.029*	0.029*	0.029*	0.057	0.343	0.200
p -value (vs SLD.)	0.886	0.057	0.686	0.486	0.686	0.200	0.029*	0.029*	0.029*
SOAM [-]	8.58 ± 0.04	8.59 ± 0.03	8.74 ± 0.02	8.75 ± 0.08	8.76 ± 0.05	8.75 ± 0.04	8.70 ± 0.05	8.57 ± 0.04	8.61 ± 0.02
p -value (vs Norm.)	0.686	0.686	0.029*	0.343	0.057	0.029*	0.057	0.686	0.343
p -value (vs SLD.)	0.200	0.200	0.114	0.886	0.886	0.057	0.029*	0.057	0.029*
EAC [a.u.]	0.77 ± 0.02	0.90 ± 0.03	0.84 ± 0.04	0.91 ± 0.03	0.94 ± 0.03	0.89 ± 0.06	0.89 ± 0.02	0.89 ± 0.03	0.85 ± 0.02
p -value (vs Norm.)	0.029*	0.057	0.686	0.114	0.029*	0.343	0.343	0.029*	0.686
p -value (vs SLD.)	0.343	0.686	0.686	0.686	0.886	0.343	0.029*	0.029*	0.029*
FDR [-]	0.68 ± 0.05	0.46 ± 0.03	0.52 ± 0.01	0.47 ± 0.03	0.44 ± 0.02	0.47 ± 0.01	0.50 ± 0.02	0.50 ± 0.02	0.52 ± 0.02
p -value (vs Norm.)	0.486	0.029*	0.057	0.200	0.029*	0.029*	0.200	0.200	0.057
p -value (vs SLD.)	0.686	0.057	0.029*	0.486	0.686	0.029*	0.114	0.029*	0.029*
Blood test	Week 0	Week 1	Week 3	Week 5	Week 7	Week 8			
ALT [U/L]	40.50 ± 4.33	31.50 ± 2.72	125.75 ± 37.88	98.25 ± 27.72	33.75 ± 1.55	31.00 ± 3.11			
p -value (vs Norm.)	0.686	0.057	0.029*	0.029*	0.657	1.000			
p -value (vs SLD.)	1.000	0.771	1.000	0.114	0.029*	0.029*			
AST [U/L]	72.25 ± 4.48	80.00 ± 8.03	173.25 ± 33.42	157.75 ± 34.81	79.75 ± 4.52	92.25 ± 11.87			
p -value (vs Norm.)	0.286	0.314	0.686	0.029*	0.543	1.000			
p -value (vs SLD.)	0.629	0.629	0.886	0.486	0.029*	0.114			
TBIL [mg/dl]	0.23 ± 0.02	0.30 ± 0.06	0.53 ± 0.17	0.35 ± 0.05	0.25 ± 0.05	0.23 ± 0.06			
p -value (vs Norm.)	1.000	0.286	0.171	0.286	0.114	0.657			
p -value (vs SLD.)	1.000	0.657	0.371	0.057	0.029*	0.029*			

Supplementary Table 5. Comparative histological quantification results between SLD recovery and SLD progression monitoring with validation experiments. Hepatic fat percentage was quantified from ORO staining, and microvascular damage was quantified by CD 31 and LYVE1 staining. *p*-values are calculated using the two-sided Wilcoxon rank-sum test, and values ≤ 0.05 are highlighted in yellow.

Histological Quantification	Fat [%]	p -value (vs Norm. week 8)	p -value (vs SLD week 8)	p -value (vs week 0)	p -value (vs weeks 1–2)	p -value (vs weeks 3–4)
Recovery (at week 8)	5.57 ± 2.78	0.029*	0.029*	0.111	0.004**	0.002**
Histological Quantification	CD31 density [%]	p -value (vs Norm. week 8)	p -value (vs SLD week 8)	p -value (vs week 0)	p -value (vs weeks 1–2)	p -value (vs weeks 3–4)
Recovery (at week 8)	30.32 ± 1.44	0.114	0.029*	0.016*	0.076	0.036*
Histological Quantification	LYVE1 density [%]	p -value (vs Norm. week 8)	p -value (vs SLD week 8)	p -value (vs week 0)	p -value (vs weeks 1–2)	p -value (vs week 4)
Recovery (at week 8)	42.78 ± 1.02	1.000	0.029*	0.686	0.029*	0.029*

For the multiparametric analyses, all indices from the groups divided according to steatosis grades were assumed to normally distributed, and analytical ROC curves were generated. This approach aligns with methods employed in previous clinical study conducting similar analyses²⁵.

17) Liver vascular anatomy and advanced statistical analyses are outside the scope of my expertise, and I am unable to fully assess these aspects of the manuscript.

Reply:

Due to the lack of literature clearly delineating each structure of the liver’s vascular anatomy, we synthesized information from multiple sources and, for clarity and consistency, adapted some of their terminology^{33, 34}. The terms hepatic artery (HA), portal vein (PV), and hepatic vein (HV) are widely recognized, and we introduced the term major hepatic vessels (MHVs) to collectively refer to these vessels. Interlobular arteries and veins (ILAVs) and sublobular veins (SLVs) are defined based on the hepatic lobule architecture, respectively representing the vessels that enter the sinusoids from the portal vein and hepatic artery, and those that exit the sinusoids and the central vein towards the hepatic vein, (please see: <https://www.elsevier.com/resources/anatomy/hepatic-lobule/page/1>). To simultaneously refer to these vessels, we introduced the term peri-lobular vessels (PLVs).

For advanced statistical analyses, please refer to the response to Comment 16.

[Suggested improvements]

Some of the comments above include suggested improvements. Additional suggestions include:

18) The reference fat percentage was obtained from histopathology, which is reasonable. However, it seems the fat percentage was directly calculated from 2D histological images without considering the partial volume effects. The fat percentage values obtained with this method does not accurately represent the percentage value in 3D. I would consider taking into account the partial volume effects.

Reply:

We acknowledge your concern regarding the limited spatial representativeness of fat percentage values derived from 2D histological sections, as they reflect only a portion of the 3D hepatic fat distribution. However, we would like to note that calculating fat percentage from 2D histological sections is a commonly employed method for diagnosing steatotic liver in clinical settings and is also widely used in preclinical studies^{3, 35-37}. To enhance the accuracy of fat quantification, we employed Oil Red O (ORO) staining instead of the conventional hematoxylin and eosin (H&E) staining, based on previous findings demonstrating that ORO provides a more consistent and objective estimation of hepatic fat content than H&E^{38, 39}.

In addition, in MCD-induced SLD rats, hepatic fat accumulation has been shown to be macro-structurally uniform across the liver. Previous study have noted that steatosis exhibits a zonal distribution, particularly concentrated in zones 2 and 3 (pericentral regions) of the hepatic lobule, indicating a microscopically localized yet macroscopically homogeneous pattern of fat deposition⁴⁰. In our study, we selected ten representative areas per slide for fat quantification, each encompassing multiple lobules to reduce sampling error. This sampling strategy allows for quantification across lobular boundaries, and therefore, the resulting fat percentage values can be regarded as a macroscopic representation of hepatic fat distribution.

*Indeed, additional ORO staining performed at SLD week 4 reveals that hepatic fat accumulation is broadly and evenly distributed (**Fig. R8a**). Five histological slides were obtained at intervals of 0.5–1 mm along the elevational direction from the fixed liver sample. A 2.5× magnified image was acquired from each slide, and five 20× magnified images were captured from one randomly selected slide. The average fat accumulation measured from the five 20× images ($42.8 \pm 1.2\%$) closely matched the average value obtained from the five 2.5× images ($46.5 \pm 4.2\%$), which were sectioned from different locations within the tissue (**Fig. R8b**).*

Fig. R8. Additional ORO staining results confirming fat distribution. **a**, Representative images at 2.5× and 20× magnification. **b**, Quantitative analysis of fat accumulation. *p*-value was calculated using Mann–Whitney U test. ORO, Oil Red O; n.s., not significant.

Therefore, we consider that the fat percentage from 2D histological sections can reasonably represent the 3D distribution of hepatic fat in this context.

19) I would encourage the authors to consider sharing the data and code.

Reply:

We would like to share a subset of the datasets (Normal week 0 and SLD week 3) and the code used in this study through a Google Drive link. Because the code embodies the core algorithmic originality and intellectual assets of this study, we kindly ask for your understanding regarding the limited availability (please refer to the following link:

<https://drive.google.com/drive/folders/1CwUrlW0rgHg9kBY3p4SBf0lSOozY5Ql9?usp=sharing>).

20) The authors mentioned “the absence of FDA-approved treatments for SLD at present” (Line 354). This statement is incorrect. FDA has approved Rezdiffra (resmetirom) for the treatment of adults with noncirrhotic metabolic dysfunction-associated steatohepatitis (MASH, previously known as non-alcoholic steatohepatitis) with moderate to advanced liver fibrosis (<https://www.fda.gov/news-events/press-announcements/fda-approves-firsttreatment-patients-liver-scarring-due-fatty-liver-disease>), and MASH is a subcategory of SLD.

Reply:

*Thank you for pointing this out. We acknowledge an inaccuracy in that section and have accordingly revised the **Discussion on Page 20**:*

“This capability would be highly valuable for testing SLD treatments, particularly given the currently limited number of FDA-approved treatments.”

21) Please consider justifying how the study time points were determined as shown in Supplementary Figure 3.

Reply:

The study time points were determined with reference to previous investigations employing the MCD diet and Wistar rat model^{3, 41}. In previous studies, fat accumulation induced by the MCD diet and its associated effects developed progressively over several weeks; therefore, a one-week interval was adopted as the standard for temporal assessment.

[Clarity and context]

22) The manuscript was well written and generally easy to follow. However, the excessive use of acronyms slowed down the reading considerably.

Reply:

*Thank you very much for your positive feedback on the manuscript and for highlighting the aspects related to improving the readability of the manuscript. To address the concern that excessive use of acronyms may hinder readability, a comprehensive list of acronyms has been provided (please refer to page 6 of this Response Letter). In addition, the table has been included as **Supplementary Table 1**, and a corresponding reference has been added in the main text.*

Main, Page 3.

*“All acronyms used in this study are listed in **Supplementary Table 1**.”*

23) The results were well presented. However, consideration of previous work was inadequate in the sense that various state-of-the-art quantitative USI techniques were not included in comparison with the proposed 3D multiparametric approach.

Reply:

Thank you for positive feedback. Please refer to the previous comments numbered 1, 2, and 3.

[References]

24) The manuscript references previous literature appropriately.

Reply:

We are glad to hear your comment.

[Code availability]

25) I highly encourage the authors to consider sharing the code.

Reply:

*Please refer to the previous comments numbered 19. The example dataset and code will be provided to potential readers upon reasonable request to the corresponding author. We have revised the **Data availability and Code availability sections**.*

Data availability, Page 34.

“The example dataset, though extensive, is available via a Google Drive link upon reasonable request to the corresponding authors.”

Code availability, Page 34.

“The example script is available via the same Google Drive link upon reasonable request to the corresponding authors.”

Reviewer #2

The study showcases 3D multiparametric ultrasound imaging as a tool for diagnosing and monitoring steatotic liver disease in rats. By integrating ultrafast Doppler imaging and quantitative USI, it quantified vascular and tissue changes over disease progression and recovery, showing strong correlation with hepatic fat accumulation. A combined US score achieved superior diagnostic accuracy, highlighting the potential of 3D USI for SLD management. However, limitations in imaging interpretation, and the diet-induced model's relevance to human disease warrant attention in interpreting and generalizing the findings. Addressing these limitations will help refine the utility of 3D UFD imaging and its application to broader experimental or clinical settings. Overall, this is an important study.

[Specific comments]

1) The methionine-choline-deficient diet induces SLD but may not fully recapitulate human SLD pathology, particularly in terms of inflammation and fibrosis. Please discuss the relevance and limitation of disease model used.

Reply:

Thank you for highlighting the limitations of the methionine-choline-deficient (MCD) diet model. Unlike human SLD, which is characterized by systemic metabolic dysfunction (obesity, insulin resistance, hyperlipidemia, etc.), MCD diet-induced SLD is confined to hepatic metabolic impairment⁴²⁻⁴⁶. This distinction constitutes a limitation in fully recapitulating the pathological features of inflammation and fibrosis observed in humans. In MCD diet-induced SLD, hepatic inflammation arises directly from lipotoxicity caused by lipid accumulation within hepatocytes, without accompanying extrahepatic abnormalities. In contrast, inflammation in human SLD results from a complex interplay of insulin resistance-associated systemic metabolic dysfunction and immune responses. Additionally, in MCD diet-induced SLD, fibrosis progresses rapidly over a short period following the onset of inflammation, typically presenting as centrilobular fibrosis. Conversely, fibrosis in human SLD develops gradually due to persistent metabolic stress and inflammatory stimuli over an extended duration, and it is characterized by more heterogeneous fibrotic patterns.

Thus, because MCD diet-induced SLD only partially reflects the metabolic etiology of human SLD, there is an inherent degree of uncertainty regarding the translational relevance of its pathological findings. However, because the observed changes primarily result from direct hepatic injury, the MCD diet model is particularly well-suited for examining liver-specific alterations^{43, 46}. Specifically, since the primary objective of our study was to investigate vascular alterations associated with hepatic fat accumulation, the MCD diet was efficient, decoupling pathological effects and enabling the observation of changes driven primarily by intrahepatic fat accumulation. This method also rapidly induces pronounced steatosis while providing consistent reproducibility of histological features^{42, 45}.

To more fully acknowledge the importance of the MCD diet's limitations in reflecting the inflammatory and fibrotic pathology of human SLD, we have revised the Discussion on Page 22:

“First, MCD diet-induced SLD only partially recapitulates human SLD pathology. Unlike human SLD, which is characterized by systemic metabolic dysfunction (obesity, insulin resistance, hyperlipidemia, etc.), MCD diet-induced SLD is confined to hepatic metabolic impairment⁴²⁻⁴⁶. This distinction introduces an inherent degree of uncertainty regarding the translational relevance of its pathological findings with regard to the inflammation and fibrosis observed in humans.”

2) Figure 2: UFD imaging provides a much clearer delineation of liver structures and vascular networks, overcoming the limitations of conventional B-mode ultrasound, which struggles to distinguish hepatic structures or major vessels. The relatively accurate measurement of vessel diameters (e.g., 175 μm and 113 μm) and the stereoscopic visualization of vascular branching, particularly the transversal connectivity of peri-lobular vessels and the ramifications from major hepatic vessels, set an advancement in hepatic vascular imaging. However, there are potential limitations to consider. The left lateral lobe was primarily targeted for 3D UFD imaging due to its stable positioning and minimal interference from other organs. This focus might limit generalizability across other liver lobes, especially regions with greater motion artifacts or proximity to other organs. How can this be resolved?

Reply:

*Thank you for underscoring this important aspect related to the generalizability of our imaging approach. Since high-quality partial 3D UFD acquisition is feasible in liver lobes beyond the left lateral lobe, generalizability to other hepatic regions can be reasonably ensured (**Fig. R9**). Image quality is partially influenced by motion artifacts and the proximity of adjacent organs, with certain regions of the median and right lateral lobes potentially affected by rib shadowing. However, even in these regions, 3D acquisition compensates for localized image quality degradation.*

Fig. R9. Volumetric 3D UFD in other liver lobes.

*To address potential concerns from readers, the above figure has been included as **Supplementary Figure 4** and cited in the main text.*

Results, Page 7.

*“The LLL was mainly targeted for 3D UFD because this region, less influenced by other organs, can be stably imaged in the elevational direction (**Supplementary Fig. 4**).”*

3) Are there any measurements fluctuations affected by respiratory or cardiac motion in live imaging of rats?

Reply:

During live imaging of rats, respiratory or cardiac motion can introduce fluctuations in the measurements; therefore, our imaging is performed with careful consideration of these physiological movements. Real time US-based respiratory gating was employed to exclude breathing phases during US acquisition. Then, from the acquired US frames, intervals with minimal cardiac motion were selected based on ensemble criteria to ensure imaging stability. For further details regarding the cardiac cycle, please refer to our response to Reviewer 1's Comment 8.

4) Figure 3: Temporal trends in hepatic fat accumulation and microvascular alterations are well-captured. Fat accumulation can alter microvascular architecture and impede blood flow within the liver. However, the precise mechanisms linking the two are not fully elucidated. Comparing Fig. 3d and 3e, vascular changes occur early by 1 week of diet, whereas changes to liver functional markers were more apparent by 3 weeks of diet. How do your findings stand in view of current literature/ evidence?

Reply:

Thank you for the insightful question. We have confirmed that the findings you mention are consistent with current literature / evidence. Although the precise mechanisms linking fat accumulation to microvascular alterations remain unclear, recent studies have demonstrated that hepatic fat accumulation in SLD can lead to microcirculatory disturbances, even in the early stages of the disease^{41, 47-51}. In particular, the impact on endothelial dysfunction and intrahepatic vascular resistance (IHVR) has been highlighted, with endothelial dysfunction caused by intrahepatic fat accumulation disrupting the balance between vasoconstriction and vasodilation, thereby contributing to elevated IHVR. Sinusoidal compression, resulting from mechanical pressure due to hepatocellular expansion caused by fat accumulation, also occurs. These functional and structural alterations have been shown to result in structural disorganization of the sinusoids^{50, 51}.

To validate the changes observed in our US findings, we performed LYVE1 staining on rats in the Normal (0 and 8 weeks), SLD (0, 2, 4, and 8 weeks), and Recovery (8 weeks) states, as recommended by you. For all groups, subjects at each time point were assumed to be in a homogeneous condition. Based on this assumption, staining was performed on four rats per time point in each group.

*In the SLD group, LYVE1 expression is reduced compared to the Normal group (**Fig. R10a**). When the LYVE1 quantification results are interpreted alongside CD31, the observed trends are consistent with those reported in the literature (**Fig. R10b and Table R4**). LYVE1 density decreases markedly at week 2, whereas CD31 levels remain relatively stable. This reduction in LYVE1 may reflect impaired endothelial function⁵²⁻⁵⁴. Conversely, at week 4, CD31 density decreases significantly, suggesting structural alterations in the microvasculature. The temporal pattern of LYVE1 and CD31 density at weeks 0, 2, and 4 suggests initial functional alterations in the sinusoids, followed by structural changes, consistent with previously reported findings^{41, 47-51}. At week 8, LYVE1 density remains relatively stable, whereas CD31 levels continue to decrease.*

This finding is further addressed in our response to Comment 5.

Fig. R10. LYVE1 and CD31 staining results from the SLD progression monitoring and validation schemes at SLD-induced weeks 0, 2, 4, and 8. **a**, Representative microscopic LYVE1 staining images from the Normal and SLD groups. **b**, Quantitative analysis of LYVE1 and CD31 densities in the Normal and SLD groups across SLD-induced weeks 0, 2, 4, and 8. All data are presented as mean \pm standard errors.

Table. R4. Quantitative comparison of LYVE1 staining from the SLD progression monitoring and validation schemes at SLD-induced weeks 0, 2, 4, and 8.

Histological quantification	Normal week 0	SLD week 2	SLD week 4	SLD week 8	Normal week 8
LYVE1 density [%]	43.68 \pm 2.31	30.61 \pm 1.60	28.16 \pm 0.61	29.88 \pm 3.00	42.62 \pm 1.28
p -value (vs Normal week 0)		0.029*	0.029*	0.029*	0.488
p -value (vs SLD week 2)			0.2	0.886	0.029*
p -value (vs SLD week 4)				0.886	0.029*
p -value (vs SLD week 8)					0.029*

With respect to liver functional markers, the literature indicates that significant changes typically emerge around weeks 3 to 4^{55, 56}. Considering that vascular alterations can occur in the early stages of SLD, our findings showing that vascular alterations emerged earlier than changes in liver functional markers are well supported by existing literature and evidence.

We consider the above-mentioned points to be highly important and have incorporated them into the main text. Revisions have been made to portions of the Introduction, Results, and Discussion sections. Please refer to the main text for the detailed changes. **Figure R10** has been incorporated into the manuscript as **Supplementary Fig. 9**, and **Table R4** has been incorporated into **Supplementary Table 3**. In addition, with the inclusion of LYVE1 staining results, the corresponding method has been added to the **Methods** section.

Introduction, Pages 2–3:

“Steatosis can cause endothelial dysfunction, disrupting the balance between vasoconstriction and vasodilation and thereby contributing to elevated intrahepatic vascular resistance. Sinusoidal compression can also occur, where hepatocytes enlarged by fatty deposits physically compress the sinusoids. Such sinusoidal changes lead to impaired microcirculation and hepatic blood flow⁵⁰.”

Results, Page 11:

“Additionally, to validate the changes observed in our USI findings, we performed LYVE1 staining on a subset of normal and SLD rats. In contrast to CD31, LYVE1 density shows a marked decline at week 2, followed by a more gradual decrease at week 4 (Supplementary Fig. 9). Given that a decrease in LYVE1 may indicate endothelial dysfunction⁵²⁻⁵⁴, the temporal patterns of LYVE1 and CD31 density at weeks 0, 2, and 4 suggest that structural alterations in the sinusoids occur subsequent to initial functional impairment. At week 8, LYVE1 levels remain comparable to those at week 4, still indicating impaired sinusoidal function, which may have contributed to the reduction in CD31 observed at week 8. Changes in UFD indices may reflect both functional and structural alterations in the microvascular architecture.”

Discussion, Page 20:

“As the induced-SLD weeks progress, the fat percentage increases, while the CD31 and LYVE1 densities decrease, suggesting that the hepatic fat accumulation leads to microvasculature disruptions. We speculate that, as demonstrated in previous studies^{41, 47-51}, this may result from endothelial dysfunction, a functional alteration occurring in the early stages of SLD that leads to structural changes in the microvasculature. Lipid droplets from hepatocytes hypertrophied by the fat accumulation physically compressing the sinusoidal structure, namely sinusoidal compression, may also have contributed to this⁵¹.”

Methods, Page 32:

“Immunofluorescence staining for CD31 and LYVE1 was performed to label vascular endothelial cells and liver sinusoidal endothelial cells, respectively.”

“Next, the sections were incubated overnight at 4°C with primary antibodies, anti-CD31 (1:200, AF3628, R&D) and anti-LYVE-1 (1:200, AF7939, R&D), and then with the secondary antibodies, rabbit anti-goat Alexa Fluor 488 (1:2000, Invitrogen) and donkey anti-sheep Alexa Fluor 488 (1:2000, Jackson ImmunoResearch), for 1 hour at room temperature.”

“Three randomly chosen images with 256×256 pixel ROI were quantified for CD31 and LYVE1 densities with ImageJ (NIH) and then averaged.”

5) Figure 4: Vascular traits appeared responsive to recovery intervention. Do vascular changes reflect genuine structural restoration or merely less compression from reduced liver fats. For

example, vascular branching by VBD, does it represent angiogenic/ active remodeling response? While the study links imaging indices to vascular and hepatic changes, additional molecular analyses in histology (e.g., angiogenic or inflammatory markers) could provide deeper physiological/ biological insights.

Reply:

We sincerely appreciate your thoughtful inquiry into areas where our work may offer novel insights. Based on our newly implemented LYVE1 staining, the observed vascular changes may indicate genuine structural restoration.

As noted in the response to Comment 4, the marked decrease in LYVE1 and CD31 density at weeks 2 and 4 suggests that structural alterations may extend to larger vascular structures beyond the sinusoids following functional changes in liver sinusoidal endothelia cells, findings that are consistent with previous studies. The reduction in UFD indices observed during the SLD progression period in our study likely reflects both functional and structural alterations within the microvascular architecture.

In the Recovery group, the microscopic image shows a significant restoration of LYVE1 expression compared to the SLD group at week 4, closely resembling that of the Normal group (Fig. R11a). Although LYVE1 density nearly returns to baseline at week 8, the persistent reduction in CD31 density suggests that the associated structural changes may be irreversible (Fig. R11b,c). The UFD indices also exhibited incomplete recovery, which may indicate that, despite functional restoration of the sinusoids, disrupted microvascular architecture hindered full recovery of in vivo blood flow.

Fig. R11. LYVE1 and CD31 staining results from the SLD recovery scheme. a, Representative LYVE1

staining images from the SLD progression, validation, and recovery schemes. **b**, Temporal quantification trends of LYVE1 and CD31 densities during SLD progression and recovery. **c**, Comparison of LYVE1 and CD31 densities between the normal and SLD recovery states. Norm, normal; and Rec, recovery.

Table R5. Comparative statistical analysis of LYVE1 density in the SLD recovery scheme.

Histological Quantification	LYVE1 density [%]	p -value vs Norm. week 8	p -value (vs SLD week 8)	p -value (vs week 0)	p -value (vs weeks 1–2)	p -value (vs week 4)
Recovery (at week 8)	42.78 ± 1.02	1.000	0.029*	0.686	0.029*	0.029*

However, interpreting the UFD indices as a direct representation of an angiogenic or active remodeling response remains challenging, as the magnitude of the changes differs, and even in the presence of neovascularization, actual blood flow can decline⁵⁷. Notably, while hemodynamic indices continued to decline, CD31 density exhibited a slight recovery trend at weeks 2 and 4 (**Fig. R12a**). Given that angiogenesis has been reported in SLD and that our results show mild inflammatory responses emerging from week 2 (**Fig. R12b**), this time point might correspond to the onset of angiogenic activity^{47, 49, 58}.

Fig. R12. CD31 quantification and H&E results from the SLD progression scheme. a, Temporal trend of CD31 density during SLD progression. **b**, Representative H&E images from Normal week 0, SLD week 2 and SLD week 4. Red arrows indicate areas of inflammatory infiltration.

Hemodynamic indices such as *wnVVO* and *wnFMBV* may be related to angiogenic activity, while vascular morphological indices such as *VBD* and *SOAM* may be associated with vascular remodeling processes. However, changes in these indices are more likely to reflect the outcomes of each vascular response rather than represent them.

In related studies^{50, 51}, rats fed an MCD diet for 4 weeks exhibited disorganized sinusoidal architecture and multiple blebs. These structures were presumed to represent closed sinusoids, compromised vascular walls with leakage, or regions of neovascularization. These findings may be associated with the changes in *VBD* and *SOAM* observed up to week 4 in SLD progression period. However, as noted earlier,

UFD indices reflect in vivo blood flow, making it challenging to establish a definitive association with the aforementioned findings. Therefore, further studies are required to validate these interpretations on a case-by-case basis.

*The above points were considered important and have been incorporated into the main text. Relevant content in the Results and Discussion sections has been revised accordingly. Given the significance of the figure, **Figure R11** has been included into the manuscript as **Supplementary Fig. 12**. **Table R4** has been incorporated into **Supplementary Table 5**.*

Results, Page 14:

*“As with SLD progression monitoring, LYVE1 staining was also performed in the recovery group for additional validation. In contrast to CD31, LYVE1 density at week 8 returned to levels comparable to those of the normal group at weeks 0 and 8 (**Supplementary Fig. 12**). This suggests that although functional recovery of the microvasculature has occurred, structural restoration remains incomplete, further supporting the possibility that these structural changes may be irreversible.”*

Discussion, Page 20:

“The UFD indices showed incomplete recovery, suggesting that overall in vivo blood flow within the hepatic vascular system may not have been fully restored. In the temporal trend of LYVE1 and CD31 density, LYVE1 returned to near-normal levels at recovery week 8, whereas CD31 levels remained below normal. This indicates that, despite functional recovery of the sinusoids, the structural alterations in the microvasculature induced during SLD progression may be irreversible.”

6) Lyve1 coupled with CD31 is a better marker for validating liver microvasculatures against your US findings. To resolve sinusoidal capillarization versus compression, can your image reconstruction discern healthy levels of fenestration versus structural alteration happening in capillarization? This would have more direct attributes in relation to what is known in human liver pathology.

Reply:

Thank you for your constructive recommendations, which have guided us in refining the analysis of our US findings. In accordance with your suggestion, we conducted additional LYVE1 staining on rats in normal, SLD, and recovery states.

The onset of sinusoidal capillarization in MCD diet-induced SLD models in rats varies across studies^{47, 49, 51, 59}. Given that the minimum resolvable scale in our images is approximately 100 μm , direct visualization of fenestrations smaller than 1 μm ⁶⁰, required to assess sinusoidal capillarization from compression, remains beyond the imaging capability. Therefore, assessing the presence of sinusoidal capillarization was beyond the scope of our study.

Because the development of sinusoidal capillarization is a key feature in the

progression of SLD, we have included a corresponding discussion in the revised manuscript. Additionally, to avoid potential confusion among readers, the reference to sinusoidal capillarization in the **Introduction** was removed to provide a more accurate description of the microvascular changes associated with SLD.

Discussion, Pages 22–23:

“Third, due to the resolution limitations ($\sim 100\ \mu\text{m}$) of our current imaging system, detecting sinusoidal capillarization, which requires visualization of fenestrations ($\sim 0.1\ \mu\text{m}$)⁶⁰, is challenging. The onset of sinusoidal capillarization in rats with SLD induced by the MCD diet varies across studies^{47, 49, 51, 59}. Sinusoidal capillarization is associated with the progression of SLD, as well as with inflammation and fibrosis, making it a key component of human SLD pathophysiology that warrants consideration in future studies.”

7) Figure 5: Diagnostic performance of 3D USI indices was benchmarked against classification grades. In the ROC curves, how do UFD indices perform compared to conventional US indices in sensitivity and specificity?

Reply:

The performance comparison between UFD indices and conventional US indices was already included, and is now presented in **Supplementary Fig. 13** (previously **Supplementary Fig. 9**, renumbered due to the addition of new supplementary figures), showing that UFD indices generally exhibit better performance than conventional US indices.

References

1. Faul F, Erdfelder E, Lang A-G, Buchner A. G* Power 3: A flexible statistical power analysis program for the social, behavioral, and biomedical sciences. *Behavior research methods*. 2007;39(2):175-91.
2. Faul F, Erdfelder E, Buchner A, Lang A-G. Statistical power analyses using G* Power 3.1: Tests for correlation and regression analyses. *Behavior research methods*. 2009;41(4):1149-60.
3. Choi C, Choi W, Kim J, Kim C. Non-invasive photothermal strain imaging of non-alcoholic fatty liver disease in live animals. *IEEE Transactions on Medical Imaging*. 2021;40(9):2487-95.
4. Bae JS, Lee DH, Lee JY, Kim H, Yu SJ, Lee J-H, et al. Assessment of hepatic steatosis by using attenuation imaging: a quantitative, easy-to-perform ultrasound technique. *European radiology*. 2019;29:6499-507.
5. Iijima H. Assessment of non-alcoholic fatty liver disease with Attenuation Imaging (ATI). Canon Med Syst Corp. 2020.
6. Son J-Y, Lee JY, Yi N-J, Lee K-W, Suh K-S, Kim KG, et al. Hepatic steatosis: assessment with acoustic structure quantification of US imaging. *Radiology*. 2016;278(1):257-64.
7. Lee DH, Lee JY, Lee KB, Han JK. Evaluation of hepatic steatosis by using acoustic structure quantification US in a rat model: comparison with pathologic examination and MR spectroscopy. *Radiology*. 2017;285(2):445-53.
8. Wear KA, Han A, Rubin JM, Gao J, Lavarello R, Cloutier G, et al. US backscatter for liver fat quantification: an AIUM-RSNA QIBA pulse-echo quantitative ultrasound initiative. *Radiology*. 2022;305(3):526-37.
9. Wang X, Bamber JC, Esquivel-Sirvent R, Ormachea J, Sidhu PS, Thomenius KE, et al. Ultrasonic sound speed estimation for liver fat quantification: a review by the AIUM-RSNA QIBA pulse-echo quantitative ultrasound initiative. *Ultrasound in Medicine & Biology*. 2023;49(11):2327-35.
10. Zhou Z, Tai D-I, Wan Y-L, Tseng J-H, Lin Y-R, Wu S, et al. Hepatic steatosis assessment with ultrasound small-window entropy imaging. *Ultrasound in medicine & biology*. 2018;44(7):1327-40.
11. Gong P, Song P, Huang C, Trzasko J, Chen S. System-independent ultrasound attenuation coefficient estimation using spectra normalization. *IEEE transactions on ultrasonics, ferroelectrics, and frequency control*. 2019;66(5):867-75.
12. Rafati I, Destremes F, Yazdani L, Gesnik M, Tang A, Cloutier G. Regularized ultrasound phantom-free local attenuation coefficient slope (ACS) imaging in homogeneous and heterogeneous tissues. *IEEE Transactions on Ultrasonics, Ferroelectrics, and Frequency Control*. 2022;69(12):3338-52.
13. Han A, Byra M, Heba E, Andre MP, Erdman Jr JW, Loomba R, et al. Noninvasive diagnosis of nonalcoholic fatty liver disease and quantification of liver fat with radiofrequency ultrasound data using one-dimensional convolutional neural networks. *Radiology*. 2020;295(2):342-50.
14. Jeon SK, Lee JM, Joo I, Yoon JH, Lee G. Two-dimensional convolutional neural network using quantitative US for noninvasive assessment of hepatic steatosis in NAFLD. *Radiology*.

2023;307(1):e221510.

15. Mitten EK, Portincasa P, Baffy G. Portal hypertension in nonalcoholic fatty liver disease: Challenges and paradigms. *Journal of Clinical and Translational Hepatology*. 2023;11(5):1201.
16. Nababan SHH, Lesmana CRA. Portal hypertension in nonalcoholic fatty liver disease: from pathogenesis to clinical practice. *Journal of clinical and translational hepatology*. 2022;10(5):979.
17. Balasubramanian P, BooPathy V, Govindasamy E, VenkatESh BP. Assessment of portal venous and hepatic artery haemodynamic variation in non-alcoholic fatty liver disease (NAFLD) patients. *Journal of clinical and diagnostic research: JCDR*. 2016;10(8):TC07.
18. Rabut C, Correia M, Finel V, Pezet S, Pernot M, Deffieux T, et al. 4D functional ultrasound imaging of whole-brain activity in rodents. *Nature methods*. 2019;16(10):994-7.
19. Bureau F, Robin J, Le Ber A, Lambert W, Fink M, Aubry A. Three-dimensional ultrasound matrix imaging. *Nature Communications*. 2023;14(1):6793.
20. Neshat H, Cool DW, Barker K, Gardi L, Kakani N, Fenster A. A 3D ultrasound scanning system for image guided liver interventions. *Medical physics*. 2013;40(11):112903.
21. Sayed A, Layne G, Abraham J, Mukdadi O. Nonlinear characterization of breast cancer using multi-compression 3D ultrasound elastography in vivo. *Ultrasonics*. 2013;53(5):979-91.
22. Debbaut C, Segers P, Cornillie P, Casteleyn C, Dierick M, Laleman W, et al. Analyzing the human liver vascular architecture by combining vascular corrosion casting and micro-CT scanning: a feasibility study. *Journal of anatomy*. 2014;224(4):509-17.
23. Masyuk TV, Ritman EL, LaRusso NF. Hepatic artery and portal vein remodeling in rat liver: vascular response to selective cholangiocyte proliferation. *The American journal of pathology*. 2003;162(4):1175-82.
24. Yoon C, Lee C, Shin K, Kim C. Motion compensation for 3D multispectral handheld photoacoustic imaging. *Biosensors*. 2022;12(12):1092.
25. Kim J, Park B, Ha J, Steinberg I, Hooper SM, Jeong C, et al. Multiparametric photoacoustic analysis of human thyroid cancers in vivo. *Cancer Research*. 2021;81(18):4849-60.
26. Chiao RY, Hao X. Coded excitation for diagnostic ultrasound: A system developer's perspective. *IEEE transactions on ultrasonics, ferroelectrics, and frequency control*. 2005;52(2):160-70.
27. Nowicki A, Secomski W, Litniewski J, Trots I, Lewin P. On the application of signal compression using Golay's codes sequences in ultrasound diagnostic. *Archives of Acoustics*. 2003;28(4).
28. Wilimitis D, Walsh CG. Practical considerations and applied examples of cross-validation for model development and evaluation in health care: tutorial. *Jmir ai*. 2023;2:e49023.
29. Allgaier J, Pryss R. Cross-validation visualized: a narrative guide to advanced methods. *Machine Learning and Knowledge Extraction*. 2024;6(2):1378-88.
30. Ng A, Swanevelde J. Resolution in ultrasound imaging. *Continuing Education in Anaesthesia, Critical Care & Pain*. 2011;11(5):186-92.
31. Pinkert MA, Hall TJ, Eliceiri KW. Challenges of conducting quantitative ultrasound with a

multimodal optical imaging system. *Physics in Medicine & Biology*. 2021;66(3):035008.

32. Drucker H, Burges CJ, Kaufman L, Smola A, Vapnik V. Support vector regression machines. *Advances in neural information processing systems*. 1996;9.
33. Teutsch HF, Schuerfeld D, Groezinger E. Three-Dimensional Reconstruction of Parenchymal Units in the Liver of the Rat. *Hepatology*. 1999;29(2):494-505.
34. Zhang S, Chen W, Zhu C. Liver Structure. *Artificial Liver*. 2021:21-47.
35. Idilman IS, Keskin O, Celik A, Savas B, Halil Elhan A, Idilman R, et al. A comparison of liver fat content as determined by magnetic resonance imaging-proton density fat fraction and MRS versus liver histology in non-alcoholic fatty liver disease. *Acta radiologica*. 2016;57(3):271-8.
36. Kleiner DE, Brunt EM, Van Natta M, Behling C, Contos MJ, Cummings OW, et al. Design and validation of a histological scoring system for nonalcoholic fatty liver disease. *Hepatology*. 2005;41(6):1313-21.
37. Carmiel-Haggai M, Cederbaum AI, Nieto N. A high-fat diet leads to the progression of non-alcoholic fatty liver disease in obese rats. *The FASEB Journal*. 2005;19(1):136-8.
38. Fiorini RN, Kirtz J, Periyasamy B, Evans Z, Haines JK, Cheng G, et al. Development of an unbiased method for the estimation of liver steatosis. *Clinical transplantation*. 2004;18(6):700-6.
39. Catta-Preta M, Mendonca LS, Fraulob-Aquino J, Aguila MB, Mandarim-de-Lacerda CA. A critical analysis of three quantitative methods of assessment of hepatic steatosis in liver biopsies. *Virchows Archiv*. 2011;459:477-85.
40. Sanches SCL, Ramalho LNZ, Augusto MJ, da Silva DM, Ramalho FS. Nonalcoholic steatohepatitis: a search for factual animal models. *BioMed research international*. 2015;2015(1):574832.
41. Francque S, Wamutu S, Chatterjee S, Van Marck E, Herman A, Ramon A, et al. Non-alcoholic steatohepatitis induces non-fibrosis-related portal hypertension associated with splanchnic vasodilation and signs of a hyperdynamic circulation in vitro and in vivo in a rat model. *Liver International*. 2010;30(3):365-75.
42. Santhekadur PK, Kumar DP, Sanyal AJ. Preclinical models of non-alcoholic fatty liver disease. *Journal of hepatology*. 2018;68(2):230-7.
43. Zhong F, Zhou X, Xu J, Gao L. Rodent models of nonalcoholic fatty liver disease. *Digestion*. 2020;101(5):522-35.
44. Carreres L, Jílková ZM, Vial G, Marche PN, Decaens T, Lerat H. Modeling diet-induced NAFLD and NASH in rats: a comprehensive review. *Biomedicines*. 2021;9(4):378.
45. Kucera O, Cervinkova Z. Experimental models of non-alcoholic fatty liver disease in rats. *World journal of gastroenterology: WJG*. 2014;20(26):8364.
46. Van Herck MA, Vonghia L, Francque SM. Animal models of nonalcoholic fatty liver disease—a starter's guide. *Nutrients*. 2017;9(10):1072.
47. Hammoutene A, Rautou P-E. Role of liver sinusoidal endothelial cells in non-alcoholic fatty liver disease. *Journal of hepatology*. 2019;70(6):1278-91.
48. Nasiri-Ansari N, Androutsakos T, Flessa C-M, Kyrou I, Siasos G, Randeve HS, et al. Endothelial

cell dysfunction and nonalcoholic fatty liver disease (NAFLD): A concise review. *Cells*. 2022;11(16):2511.

49. Furuta K, Guo Q, Hirsova P, Ibrahim SH. Emerging roles of liver sinusoidal endothelial cells in nonalcoholic steatohepatitis. *Biology*. 2020;9(11):395.

50. Francque S, Laleman W, Verbeke L, Van Steenkiste C, Casteleyn C, Kwanten W, et al. Increased intrahepatic resistance in severe steatosis: endothelial dysfunction, vasoconstrictor overproduction and altered microvascular architecture. *Laboratory investigation*. 2012;92(10):1428-39.

51. van der Graaff D, Chotkoe S, De Winter B, De Man J, Casteleyn C, Timmermans J-P, et al. Vasoconstrictor antagonism improves functional and structural vascular alterations and liver damage in rats with early NAFLD. *JHEP Reports*. 2022;4(2):100412.

52. Gage BK, Liu JC, Innes BT, MacParland SA, McGilvray ID, Bader GD, et al. Generation of functional liver sinusoidal endothelial cells from human pluripotent stem-cell-derived venous angioblasts. *Cell Stem Cell*. 2020;27(2):254-69. e9.

53. Arimoto J, Ikura Y, Suekane T, Nakagawa M, Kitabayashi C, Iwasa Y, et al. Expression of LYVE-1 in sinusoidal endothelium is reduced in chronically inflamed human livers. *Journal of gastroenterology*. 2010;45:317-25.

54. Lin Y, Dong MQ, Liu ZM, Xu M, Huang ZH, Liu HJ, et al. A strategy of vascular-targeted therapy for liver fibrosis. *Hepatology*. 2022;76(3):660-75.

55. Veteläinen R, Van Vliet A, Van Gulik TM. Essential pathogenic and metabolic differences in steatosis induced by choline or methionine-choline deficient diets in a rat model. *Journal of gastroenterology and hepatology*. 2007;22(9):1526-33.

56. Palladini G, Di Pasqua LG, Cagna M, Croce AC, Perlini S, Mannucci B, et al. MCD diet rat model induces alterations in zinc and iron during NAFLD progression from steatosis to steatohepatitis. *International Journal of Molecular Sciences*. 2022;23(12):6817.

57. Zhang W, Huang C, Yin T, Miao X, Deng H, Zheng R, et al. Ultrasensitive US microvessel imaging of hepatic microcirculation in the cirrhotic rat liver. *Radiology*. 2022;307(1):e220739.

58. Lei L, El Mourabit H, Housset C, Cadoret A, Lemoine S. Role of Angiogenesis in the Pathogenesis of NAFLD. *Journal of clinical medicine*. 2021;10(7):1338.

59. Rautou P-E, Chotkoe S, Biquard L, Wettstein G, Van der Graaff D, Liu Y, et al. Altered liver sinusoidal endothelial cells in MASLD and their evolution following lanifibranor treatment. *JHEP Reports*. 2025:101366.

60. Li W, Lv J, Li H, Song L, Zhang R, Zhao X, et al. Quantification of Vascular Remodeling and Sinusoidal Capillarization to Assess Liver Fibrosis with Photoacoustic Imaging. *Radiology*. 2025;314(1):e241275.

Reviewer #1

The authors have done an excellent job addressing the reviewers' comments, and the quality of the work has been significantly improved after the revision. A few additional issues were noted:

Reply:

We sincerely appreciate your favorable comments and the constructive comments and suggestions you previously provided. We also believe that our work has been significantly improved through the previous revision. Please find our detailed responses to the additional issues you raised below.

1) The issue of excessive use of acronyms has not been resolved. I would highly recommend considering reducing the use of acronyms in this manuscript to improve readability.

Reply:

We have carefully reviewed the manuscript and reduced the use of acronyms to improve readability, as you recommended. In particular, we have revised liver structural and vascular anatomy terminology by replacing acronyms (e.g., 'LLL' and 'HAs') with full terms (e.g., 'left lateral lobe' and 'hepatic arteries'). Additionally, we have replaced infrequently used acronyms (e.g., PRF) with their full terms (e.g., pulse repetition frequency) without abbreviations. Please note that acronyms have been retained in the figures due to space constraints, and we appreciate your understanding.

2) Line 421: "speed of sound estimation requires specialized acquisition sequences": The same statement is true for ultrafast Doppler.

Reply:

We agree that the statement holds true for ultrafast Doppler. However, we would like to emphasize that spatially uncompounded data acquired at multiple steering angles is required for speed of sound estimation. This setting substantially increases storage demands, making it impractical for our experimental conditions, which involve a large number of volumetric data acquisitions.

*The **Discussion section on Page 22** has been revised to clarify the intent of the statement:*

“Specifically, backscatter coefficient estimation requires precise calibration with reference phantoms, and speed of sound estimation requires specialized acquisition sequences for spatially uncompounded data at multiple steering angles, both of which were challenging under our experimental condition.”

3) Line 422: "obtaining accurate estimates using these two techniques necessitates access to raw channel data": This statement is incorrect. Backscatter coefficient measurement does not require access to channel data. Some of the speed of sound methods do not require access to channel data either.

Reply:

Thank you for bringing this inaccuracy to our attention. We have thoroughly reviewed the information on the backscatter coefficient and speed of sound measurement, and have revised the channel data access statement accordingly.

*The **Discussion section on Page 22** has been updated to correct inaccuracies in the statement:*

“Furthermore, improving the accuracy of the speed of sound estimation technique requires access to raw channel data, which is generally unavailable in clinical US systems and, when combined with the aforementioned requirements, substantially increases computational

complexity.”

4) Line 470: "thereby enabling the use of linear transducers with higher center frequencies": I am unsure if the use of linear transducers would be appropriate for liver imaging, given the limited field of view of linear transducers.

Reply:

*We agree that your concerns about the applicability of linear transducers in human liver imaging are valid. However, despite their limited field of view, linear-array transducers provide a more detailed depiction of vessel distribution than convex-array transducers due to their higher center frequency, which may render them useful in human liver imaging. As in our previous revision, we employed the Vantage 256 system and obtained human liver images using the GE 9L-D probe (**Fig. S1**). UFD Images were acquired via the intercostal window at a frame rate of 300 Hz with an ensemble length of 240 frames (0.8 s). As shown in the figure, the linear-array transducer enables acquisition of higher-resolution vascular images of the human liver.*

Fig. SR1. Human liver UFD images from a healthy volunteer (male, 23 years old). **a**, Cross-sectional PD image obtained using convex-array US probe (GE C1-6-D). **b**, Cross-sectional PD image obtained using linear-array US probe (GE 9L-D). **c**, Magnified views of (a) and (b). Vantage 256 system was employed for imaging. 400 frames (500 Hz frame rate) and 240 frames (300 Hz frame rate) were acquired via the intercostal window using convex-array and linear-array probes, respectively.

To further emphasize the potential of human liver UFD, Figure SR1b has been included as Supplementary Figure 19b, and the Discussion section on Page 23 has been revised:

“Extending our research to extensive clinical studies could address these limitations. In practice, we are currently preparing an extended study on human liver

imaging, utilizing the Vantage 256 system along with a convex-array (GE C1-6-D) and linear-array (GE 9L-D) probes to acquire high-quality human liver images (Supplementary Fig. 19a,b). A total of 400 frames (500 Hz frame rate) and 240 frames (300 Hz frame rate) were acquired via the intercostal window using the convex-array and linear-array probes, respectively, from a single healthy volunteer (male, 23 years old). Ensemble frames for UFD were selected at 0.8-s intervals, considering the human cardiac cycle and resting heart rate (60–100 bpm). As shown in the figures, the human hepatic vasculature is visualized in sufficient detail across an extensive region with the convex-array probe, and in finer detail within a narrower region with the linear-array probe.”

5) In response to the reviewer's question about the sample size, the authors added a post hoc power analysis to the revised manuscript. The value of this post hoc power analysis is unclear. As a general guideline, post hoc power analysis should not be used, as explained in the following publication: Heinsberg LW, Weeks DE. Post hoc power is not informative. *Genet Epidemiol.* 2022 Oct;46(7):390-394.

Reply:

We reviewed the post hoc power analysis using the reference you provided, and we fully agree with your comment. Post hoc power analysis provide no additional value for interpreting the results and may even cause misunderstanding. We have excluded the post hoc power analysis previously included in Supplementary Note 2 from the manuscript.

Reviewer #2

My comments have been adequately addressed.

Reply:

We sincerely appreciate your earlier insightful comments and suggestions, through which our work has been considerably strengthened.

Comments on Manuscript# NCOMMS-24-84492

Key results

In this manuscript, the authors demonstrated that 3D multiparametric hepatic ultrasound imaging (USI) could be a useful modality for diagnosis and monitoring of steatotic liver disease (SLD). They performed an animal study on 37 rats that underwent 3D ultrasound liver scans in vivo. The 3D multiparametric USI method used by the authors involves the combination of two well-established USI techniques, ultrafast Doppler imaging (UFD) and quantitative USI. UFD provides visualization of the liver microvasculature, and quantitative USI provides information about acoustic properties of the liver. It is well known in the literature that quantitative USI parameters such as the attenuation coefficient and acoustic structure quantification are correlated with liver fat content, and this study added knowledge to the literature by showing that UFD-derived parameters were also correlated with liver fat content and might have superior diagnostic performance compared with quantitative USI parameters (i.e., tissue-derived US indices).

Validity of conclusions

The conclusion that “3D multiparametric USI can be an invaluable modality for diagnosis/monitoring of SLD” is well supported by the data presented in the manuscript. However, the claimed superiority of UFD relative to quantitative USI is not adequately supported, for several reasons:

1) The authors only examined two quantitative USI parameters, the attenuation coefficient and the acoustic structure quantification. Other quantitative USI parameters that are known to strongly correlate with liver steatosis, such as the backscatter coefficient (<https://doi.org/10.1148/radiol.220606>), speed of sound (<https://doi.org/10.1016/j.ultrasmedbio.2023.06.021>), and several envelope statistics parameters, are not included in the comparison. Also, the rationale for including only the quantitative USI parameters (attenuation coefficient and the acoustic structure quantification) and excluding the other parameters was not provided.

2) The attenuation coefficient algorithm implemented in this study does not represent state of the art, which could lead to an underestimate of the performance of attenuation coefficient. The authors are encouraged to use state-of-the-art attenuation algorithms (e.g., <https://doi.org/10.1109/TUFFC.2017.2719962>, <https://doi.org/10.1109/TUFFC.2019.2903010>, <https://doi.org/10.1109/TUFFC.2022.3218920>) for a fairer comparison.

3) A wide body of literature is available on artificial intelligence (AI)-based liver fat quantification using quantitative USI, which often yields highly accurate results. The

superiority of UFD cannot be adequately supported without comparison with state-of-the-art AI-based quantitative USI.

Significance

1) The study is significant in that it generates new knowledge that UFD-derived parameters could be highly correlated with liver fat. However, the significance in this aspect is slightly compromised by the fact that several studies have demonstrated the usefulness of conventional Doppler for SLD diagnosis and monitoring

(<https://doi.org/10.1016/j.aohep.2020.09.008>). If conventional Doppler is useful, it is expected that UFD could be better than conventional Doppler. Prior studies have also reported that severity of fatty liver is associated with liver vascular injuries (<https://doi.org/10.1016/j.aohep.2020.09.008>), although the current study is more comprehensive.

2) Technical novelty of the manuscript seems limited, as UFD and quantitative USI are well-established methods and have previously been used in 3D as well.

3) The potential clinical impact of the study is uncertain given the potential challenges associated with performing UFD scans in humans, as the authors acknowledged. The authors mentioned several challenges, including the need for accelerated data processing (the current total imaging time is 10-12 minutes) and the challenges in obtaining 3D images from the human abdomen. While the challenge in imaging time is easier to address, the challenge in 3D imaging is more difficult to overcome due to the limited acoustic window size for liver imaging, particularly when an intercostal approach is used for liver imaging.

Further, there are several fundamental challenges that the authors did not discuss that may significantly impact clinical translation. The impressive UFD images presented in the manuscript were acquired at a center frequency of 18 MHz, with a pulse repetition frequency of 20 kHz and an imaging depth of a few millimeters. For clinical human liver imaging, however, the center frequency is typically around 3 MHz. The significant reduction in the ultrasound frequency will lead to not only significantly worse image resolution, but also significantly lower sensitivity to blood flow, both of which lead to challenges in extrapolating the performance of UFD from rats to humans. In addition, the drastically increased imaging depth required in human liver imaging will lead to a reduction of pulse repetition frequency, which will adversely affect UFD image quality.

4) Existing quantitative USI methods have already shown good accuracy in liver fat quantification. The added value of UFD is unclear if the superiority of UFD is not adequately supported. More importantly, primary challenges with some of the existing USI methods are their lack of reproducibility, as shown in <https://doi.org/10.1148/radiol.240162>. The

reproducibility of UFD was not addressed in the manuscript, which casts further doubts in its added value. UFD-derived parameters are dependent on numerous signal and image processing parameters such as the singular value decomposition cutoff values and vascular modeling processes. Therefore, it seems crucial to address the reproducibility of UFD-derived parameters in this study to demonstrate inter-operator and inter-system agreement.

Data and methodology

Overall the animal study design and the UFD data acquisition and processing are solid, and the presentation of the manuscript is reasonable. However, a few issues and potential flaws are noted, and discussed as follows.

- 1)** The authors developed a clever method for automatic ensemble selection during UFD processing (Supplemental Figure 1b), which is excellent. However, it appears that the cardiac cycle was not considered in their data analysis. Blood flow is sensitive to the cardiac cycle, which indicates that UFD-derived parameters may be dependent on the cardiac cycle. Ignoring the cardiac cycle may lead to increased variability in the UFD-derived parameters.
- 2)** The UFD data acquisition workflow could be optimized to allow faster imaging. First example, instead of waiting until all acquired data have been beamformed to move to the next scan location, the authors could have acquired all the data first and performed beamformed through post processing, which would significantly reduce the data acquisition time.
- 3)** There appears to be a flaw in how the “all-around US score” was derived. The authors “tested all possible combinations of the six US indices, a total of 63 cases (Supplementary Table. 3)”, which should be viewed as a feature selection process in machine learning terminology. This feature selection process should be performed using a separate dataset, which was not the case in this manuscript. Even though the authors performed cross-validation to evaluate the performance and check overfitting, the cross-validation process performed by the authors did not involve feature selection, and therefore potential bias in feature selection could still lead to an over-estimation of the performance.
- 4)** Figure 3-g showed a moderate correlation (Pearson correlation coefficient: -0.71; Spearman’s correlation coefficient, -0.63) between CD31 density (direct quantification of microvascular damage) and fat percentage, and the correlation was weaker than those between the UFD-derived parameters and fat percentage (please see “Hemodynamics” and “Vascular morphology” plots in Figure 5-a). Given that the CD31 density is a direct quantification of microvascular damage, and UFD-derived parameters are only surrogate

measures of microvascular damage, can the authors provide insight into why the UFD-derived parameters showed a higher correlation with fat percentage than CD31 density?

5) Lines 392-395: Can the authors clarify how a pulse repetition frequency of 20 kHz combined with 11-angle planewave pulses resulted in a B-mode frame rate of 1000 Hz? The math did not seem right, or I might be missing something here.

6) Line 396: The step size was 0.2 mm in the elevational direction of the transducer, which was 1/8 of the transducer's elevational width of 1.6 mm (<https://verasonics.com/high-frequency-transducers/>). Can the authors justify the use of such a small step size?

7) Line 448: “the window size along the depth direction and the overlap length were optimized experimentally” – can the authors clarify the optimization process? This detail is important to ensure transparency.

8) Can the authors clarify why support vector regression (SVR), as opposed to other methods, was used to combine the ultrasound indices?

9) The sample size does not seem to be adequately justified. The sample size of 37 rats seems small compared to the types of statistical analyses performed in the study; however, it is beyond the scope of my expertise to scientifically assess the appropriateness of this sample size.

10) Liver vascular anatomy and advanced statistical analyses are outside the scope of my expertise, and I am unable to fully assess these aspects of the manuscript.

Suggested improvements

Some of the comments above include suggested improvements. Additional suggestions include:

1) The reference fat percentage was obtained from histopathology, which is reasonable. However, it seems the fat percentage was directly calculated from 2D histological images without considering the partial volume effects. The fat percentage values obtained with this method does not accurately represent the percentage value in 3D. I would consider taking into account the partial volume effects.

2) I would encourage the authors to consider sharing the data and code.

3) The authors mentioned “the absence of FDA-approved treatments for SLD at present” (Line 354). This statement is incorrect. FDA has approved Rezdiffra (resmetirom) for the treatment of adults with noncirrhotic metabolic dysfunction-associated steatohepatitis (MASH, previously known as non-alcoholic steatohepatitis) with moderate to advanced liver fibrosis (<https://www.fda.gov/news-events/press-announcements/fda-approves-first->

treatment-patients-liver-scarring-due-fatty-liver-disease), and MASH is a subcategory of SLD.

4) Please consider justifying how the study time points were determined as shown in Supplementary Figure 3.

Clarity and context

The manuscript was well written and generally easy to follow. However, the excessive use of acronyms slowed down the reading considerably.

The results were well presented. However, consideration of previous work was inadequate in the sense that various state-of-the-art quantitative USI techniques were not included in comparison with the proposed 3D multiparametric approach.

References

The manuscript references previous literature appropriately.